# Electrochemical Nanobiosensors for Detection of Breast Cancer Biomarkers

**DOI:** 10.3390/s20144022

**Published:** 2020-07-20

**Authors:** Veronika Gajdosova, Lenka Lorencova, Peter Kasak, Jan Tkac

**Affiliations:** 1Institute of Chemistry, Slovak Academy of Sciences, Dubravska cesta 9, 845 38 Bratislava, Slovakia; Veronika.Gajdosova@savba.sk (V.G.); Lenka.Lorencova@savba.sk (L.L.); 2Center for Advanced Materials, Qatar University, Doha 2713, Qatar

**Keywords:** breast cancer, nanobiosensors, biomarkers, electrochemistry, impedance, immobilization, nanomaterial, nanoparticles (NPs), magnetic NPs, self-assembled monolayers (SAMs), signal amplification

## Abstract

This comprehensive review paper describes recent advances made in the field of electrochemical nanobiosensors for the detection of breast cancer (BC) biomarkers such as specific genes, microRNA, proteins, circulating tumor cells, BC cell lines, and exosomes or exosome-derived biomarkers. Besides the description of key functional characteristics of electrochemical nanobiosensors, the reader can find basic statistic information about BC incidence and mortality, breast pathology, and current clinically used BC biomarkers. The final part of the review is focused on challenges that need to be addressed in order to apply electrochemical nanobiosensors in a clinical practice.

## 1. Introduction

According to the World Health Organization, the year of 2030 should witness roughly 12 million cancer-related deaths, making cancer one of the most prominent death-causing factors around the globe. In fact, the number of new cases of cancer (cancer incidence) is currently around 439 *per* 100,000 men and women *per* year [1]. Breast cancer (BC) has been considered the most frequent type of cancer disease worldwide among women, impacting 2.1 million women each year. In 2018, it was estimated that around 627,000 women died from BC; that is approximately 15% of all cancer deaths among women [2]. The highest incidence rates were observed in the United States and Western Europe. In the US, there were 101 new cases reported *per* 100,000 women, and in Europe, there were 85 [3]. East Asia has the lowest incidence with 21 cases *per* 100,000 women [3]. In Africa, the incidence is slightly higher with 23 cases *per* 100,000 women, but this amount can be undervalued due to a lack of accurate data [3].

BC is one of the leading causes of cancer-related mortality. The disease had always been common among women. That is supported by the fact that one of the first surgical treatments ever performed was BC treatment during the first surgical revolution at the end of the 19th century. BC rates are globally increasing and are higher among women in developed regions. The incidence and fatality increase with the increasing age of women as well. It was reported that statistically, women with an age 65 and above die with higher probability due to the disease [4,5,6]. The probability of the disease to develop within a woman’s lifetime has grown over the past few decades from 1 in 11 in 1975 to 1 in 8 in 2016 [6]. There are several risk factors behind BC, including age, geographic location (country of origin), socioeconomic status, lifestyle risk factors (smoking, alcohol, diet, obesity, and physical activity), low rates of breastfeeding, family history of BC, mammographic density, ionizing radiation, etc. [5].

If the BC is diagnosed at an early stage, a 5-year survival rate can reach up to 90% in developed countries [7]. On the other hand, once a BC is metastatic, the patient’s 5-year survival rate falls down to 27.4% [8]. Early diagnosis is needed for a successful treatment and high survival rate. T1 tumors with size less than 2 cm show a 10-year survival rate of approximately 85%, while T3 tumors show a 10-year survival of less than 60% mainly as the result of delayed accurate diagnosis [9]. Nowadays, mammography is used as a gold standard for early BC screening and detection, but it is less sensitive for young women (under 40 years old) with a sensitivity of 25–59%. A factor that is limiting the diagnosis of young women is a denser breast tissue compared to older women. Other limitations of mammography are high rates of false-positive and false-negative results which lead to biopsy, high cost of treatment, and procedural discomfort for the women [10].

To avoid development of the disease into advanced stages, there is clear need for early diagnostics, efficient treatment, and post-treatment monitoring. Therefore, there is an enormous demand for efficient less-invasive diagnosis i.e., analysis of cancer biomarkers in plasma/serum samples [11].

Although several review papers have been published recently describing the electrochemical biosensing of cancer biomarkers [12,13,14,15,16], such studies only partly covered the biosensing of BC biomarkers or the electrochemical biosensing of BC biomarkers. There are only two review papers specifically covering the electrochemical biosensing of BC biomarkers published in 2017 [17,18], but with only a minor coverage of beneficial properties of nanoparticles within electrochemical transducing schemes. To our best knowledge, this is the first review paper comprehensively covering the use of nanomaterials for enhanced electrochemical detection of breast cancer biomarkers.

## 2. Breast Pathology

In humans, the breast has a number of functions. The mammary gland is a distinguishing feature of mammals, and its primary role is to produce milk to nourish offspring. The breast develops in the superficial fascia. Until puberty, the breast includes only few ducts in both men and women. In females, true breast development begins at puberty due to the effect of estrogen and progesterone [19]. The breast consists of 12–15 major breast ducts, which lead to the formation of a nipple. These are connected to ducts ending in a duct lobular unit, which is the functional milk-producing unit of a breast [20]. Breast ducts are surrounded by myoepithelial cells supported by connective tissue stroma and a variable amount of fat [19]. The terminal duct lobular units spread during pregnancy. Milk is produced due to the secretion of prolactin and oxytocin. Imbalance in estrogen and progesterone concentrations *prior* to menopause results in atrophic changes of a breast tissue.

From a clinical point of view, the lymphatic drainage of a breast has a big importance. Approximately 5% of the lymph from the breast drains through the intercostal spaces to nodes along the internal mammary vessels. The remaining 95% of the lymph drains toward the axilla in one or two larger channels. As a result of this fact, all patients with invasive BC should go through some form of auxiliary surgery to find out whether there is lymph node involvement [21].

BC is a heterogeneous disease with various subtypes accompanied by a series of genetic changes. Breast tumors originate in the anatomical structures of the mammary gland that form the mammary gland, fibrous tissue, and adipose tissue. Mammary gland tumors can be benign such as papillomas and fibroadenomas. The most common malignant tumors are carcinomas. BC is most often caused by the terminal lobes (lobular) of the mammary gland and their ducts (ductal) (Figure 1). Globally, approximately 80% of all diagnosed BC cases are of the ductal subtype [22]. These two subtypes cover 40%–75% of all diagnosed cases [23]. Other types of cancer are present in 10% of cases where inflammatory BC, male BC, Paget’s disease of breast, papillary carcinoma, and others are included [24]. BC can be classified by immunohistochemical examination into four subtypes: estrogen receptor (ER), progesterone receptor (PR), human receptor tyrosine-protein kinase erbB-2 (HER2), and antigen Ki-67 dependent [25].

## 3. BC Biomarkers

Due to progress in genomics, proteomics, and glycomics, various candidate biomarkers have been identified with a clinical potential for BC management [27]. A tumor marker was first discovered in 1847, and currently, there are more than 100 known different tumor markers [28]. Biomarkers have great potential for screening and diagnostics because they are present in blood and provide information about the health condition [7]. In healthy individuals, the tumor marker concentration is at a very low level or even in some cases absent, while increased values can reveal development and/or progression of a disease [29]. Serum biomarkers providing key information about the disease are important for the management of cancer patients, since blood aspiration is only a moderately invasive procedure.

The most relevant biomarkers that have occurred in BC include the presence of gene markers such as BReast CAncer Type (*BRCA1*, *BRCA2*), and protein-based biomarkers including cancer antigen CA 27.29, carcinoembryonic antigen (CEA), human epidermal growth factor receptor 2 (HER2), vascular endothelial growth factor (VEGF), polypeptide antigen (TPA), cytokeratin 19 fragment (CIFRA-21-1), platelet-derived growth factor (PDGF), and osteopontin (OPN). The basic characteristics of BC biomarkers are summarized in Table 1.

BC biomarkers (glycoproteins: mucin 1 (MUC1), HER2, carcinoembryonic antigen (CEA), epidermal growth factor receptor (EGFR), carbohydrate antigen 15-3 (CA15-3), CA 27-29, mammaglobin (MAM); DNA: *BRCA1*, *BRCA2*, proteins: Ki-67, OPN, microRNAs, and circulating tumor cells (CTC)) can be classified as diagnostic (healthy versus BC), prognostic (early BC versus advanced BC), predictive (provide information regarding whether a particular treatment will be beneficial for the BC patient) or therapeutic (a target biomolecule for therapeutics) based biomarkers [30,31].

Interestingly, although many studies have been published, only 9 biomarkers of cancer have been approved by the Food and Drug Administration (FDA) for clinical examinations so far. Since these biomarkers are all glycosylated proteins, changes in the glycan composition of these glycoproteins may serve as additional information for cancer diagnostics and/or prognosis. The following glycoprotein-based biomarkers have been published in the literature for BC management: HER2/NEU, CA15-3, CA27.29, MAM, galectin 3 binding protein, nectin 4, and fibronectin 1 with a typical concentration in human serum of 1–50 ng/mL [30]. The invasion, migration, and metastasis of cancer are caused by the deregulation of glycosylation related to an essential post-translational modification of proteins. In more than 90% of BC cases, there are observed changes in *O*-linked mucin-type glycosylation for example, expression of the Tn antigen, and the loss of core 2 *O*-glycans [32]. In BC, major glycan changes involve increased sialylation and fucosylation [33,34,35,36].

When new BC biomarkers are identified, it is of high importance to compare their clinical performance with already approved biomarkers in a form of AUC (area under curve) values in the receiver operating characteristic (ROC). The ROC curves calculated using the Youden index for the determination of CEA and CA15-3 showed AUC values of 0.616 or 0.678, respectively for CEA (cut-off value of 3.2 ng/mL) and CA15-3 (cut-off value of 13.3 ng/mL), when applied for disease-free survival examination [37]. We have found out clinical parameters for some of the novel BC biomarkers, showing significant advantage over already approved biomarkers. For example, Park et al. found out that the level of the human cytosolic thioredoxin correlated very well with the progress of BC [38]. At the cut-off value of 33.2 ng/mL, a sensitivity of 89.8%, a specificity of 78.0%, and an AUC value of 0.901 ± 0.025 were obtained by applying the ELISA format for thioredoxin analysis [38]. The ELISA method developed by Bernstein et al. resulted in an estimated AUC value of 0.892 using mammaglobin detection [39]. Thus, their method was able to distinguish healthy women from those having BC with high accuracy [39]. Yan et al. examined a clinical potential of the level of one type of fucosyltransferase (FUT4) determined in blood serum by ELISA for BC diagnostics [40]. AUC values of 0.784, 0.468, and 0.563 were determined for FUT4, CA15-3, and CEA, respectively. The results pointed out that FUT4 is in correlation with CA15-3 (*p* < 0.05). Moreover, FUT4 could be applied besides BC diagnostics also for BC prognosis [40].

## 4. Nanomaterials/Nanoparticles-Based Electrochemical Biosensors as Ultrasensitive Tools in Detection of BC Biomarkers

The speech of the physicist Richard Feynman entitled “There’s plenty of room at the bottom”, which took place at the Meeting of the American Physical Society in 1959 at CalTech, is considered to be the beginning of the nanotechnology era. Significant attention is currently being paid to nanomaterials. Nanomaterials are considered a pivotal tool for numerous applications in part due to their high surface area, compared to their respective bulk forms. Nanostructures with at least one dimension of size of 100 nm (1 nm = 1 × 10^−9^ m) or smaller are extremely useful in a number of areas, such as electronics, aerospace, military, pharmaceuticals, medicine, etc. Within last years, there has been an improvement in the synthesis and characterization of different nanomaterials, such as carbon-based nanomaterials, hydrogels, magnetic nanoparticles, metallic nanoparticles, polymer nanoparticles, and/or nanocomposites and two-dimensional nanomaterials [41,42].

One of the leading areas for practical application of the state-of-the-art nanoscience and nanotechnology is the development of various types of biosensors.

The application of nanomaterials to design biosensing platforms offers exceptional electronic, magnetic, mechanical, and optical properties for such devices. Nanomaterials can increase the surface of the transducing area of the sensors, which in turn provides enhanced catalytic activity. Electroactive properties of nanoparticles toward certain reactions have been widely exploited in biosensing applications. Nanometer-size structures have a large surface-to-volume ratio, controlled morphology, and structure that would scale down the characteristic size, which is a clear advantage when the sample volume is critical. The integration of advanced 2D nanomaterial MXene into biosensors architecture brings the advantage of hydrophilic character due to functional groups onto the nanoscale surface [43]. However, advances in nanomaterial biofunctionalization are crucial to achieve higher specificity in biosensing. To that end, nanomaterials can be “decorated” with different (bio)receptors offering specific recognition for biosensing [44,45,46,47,48,49]. There are basically two approaches applied to designing nanobiosensors: i.e., the application of nanoparticles for the modification of electrode surfaces (Approach 1, Figure 2), or the application of nanoparticles to make signal nanoprobes that enhance a generated signal (Approach 2, Figure 2). There are some nanobiosensors constructed using both amplification approaches (hybrid biosensing, i.e., Approach 3). In the forthcoming sections, when discussing particular nanobiosensors, amplification strategies are indicated as well.

Electrochemical biosensors (amperometric, potentiometric, conductometric, impedimetric, field-effect devices, etc.) are of particular interest for early-stage diagnostics of cancer diseases [15,50]. Electrochemical techniques such as cyclic voltammetry (CV), chronoamperometry (CA), differential pulse voltammetry (DPV), electrochemical impedance spectroscopy (EIS), and square wave voltammetry (SWV) offer an easy-to-use, affordable, highly sensitive, and reliable way for the ultrasensitive sensing of biomarkers related to such diseases [51,52]. Lab-on-chip biosensors presenting the compact and low-power portable miniaturized devices can be utilized in cancer biomarker discovery research, leading to potential clinical applications [53,54,55,56,57]. The surface architecture connecting the sensing element to the biological sample at the nanometer scale determines signal transduction and the general performance of electrochemical sensors. The eventual biosensor sensitivity is affected by the most common surface modification techniques with subsequent functionalization, various electrochemical transduction mechanisms, and by the choice of the recognition element (antibodies, nucleic acids, cells, micro-organisms, etc.). Electrochemical biosensors employing surface nanoarchitectures offer attractive features including robustness, easy miniaturization, excellent detection limits, as well as small analyte volumes and the ability to be used in turbid biofluids with optically absorbing and fluorescing compounds.

Nonetheless, there is still great room for improvement with regard to reproducibility, specificity, stability, and assay throughput of biosensing assay formats.

Regarding the sensitivity of detection by the biosensors, there is need to achieve a limit of detection (LOD) for the analytes that is at least comparable with ELISA assay format offering LODs of 0.75 ng/mL (HER2), 0.1 μg/L (kallikrein 5), and 0.17 ng/mL (thymidine kinase (TK1)). It is also important to outperform ELISA by the design of electrochemical biosensors offering to complete the whole assay procedure within 5 h and with a moderate throughput of analysis (up to 50 samples analyzed *per* run), which is typical for ELISA-based assay formats [58,59,60]. The novel generation of highly specific, sensitive, selective, and reliable micro (bio-)chemical sensors and sensor arrays can merge interdisciplinary knowledge in bio- and electrochemistry, solid-state chemistry, surface physics, bioengineering, integrated circuit silicon technology, and data processing. In the forthcoming sections, we discuss novel, nanoparticle-based approaches for the electrochemical detection of BC biomarkers.

### 4.1. Detection of DNAs

The product of a BReast CAncer Type 1 (*BRCA1*) gene controls the cell cycle and ensures DNA repair. Mutation in the *BRCA1* gene leads to BC predisposition due to the loss of a gene function [61].

Benvidi and co-workers are topically focused on the development of DNA biosensors for the detection of *BRCA1* mutation at initial stages [62,63,64,65]. Benvidi and Jahanbani [63] applied a carbon paste electrode in combination with metallic nanocomposite, i.e., a magnetic bar carbon paste electrode decorated with magnetic iron oxide and silver nanoparticles by a physical method for label-free DNA detection. In the next step, the nanocomposite was modified with a self-assembled monolayer (SAM) of thiolated single-stranded DNA. The biosensor detected *BRCA1* 5382 mutation by the EIS method with an LOD of 3.0 × 10^−17^ M (a linear range from 1.0 × 10^−16^ M to 1.0 × 10^−8^ M) [63] (Approach 1). Benvidi et al. [64] published results obtained with an improved strategy based on the application of glassy carbon electrode (GCE) modified by another type of carbon nanomaterial, e.g., reduced graphene oxide (RGO) or multi-walled carbon nanotubes (MWCNTs). 1-pyrenebutyric acid-*N*-hydroxysuccinimide ester was applied as a scaffold molecule for the immobilization of a *BRCA1* DNA probe for the detection of complementary DNA sequences. By applying this approach, the authors obtained a low LOD of 3.1 × 10^−18^ M and 3.5 × 10^−19^ M for a MWCNT-modified or RGO-modified device, respectively [64] (Approach 1). Other work from the same group of authors [65] was focused on an advanced coating of GCE with a dispersion of GO and a silk fibroin (SF) with subsequently electrochemically immobilized gold nanoparticles (AuNPs) for *BRCA1* 5382 mutation detection. In the next step, the analyte was incubated with the modified electrode and measured by CV and EIS techniques with complementary target DNA sequences. The impedimetric DNA sensor achieved an LOD of 3.3 × 10^−17^ M (a linear range from 1.0 × 10^−16^ M to 1.0 × 10^−8^ M) [65] (Approach 1).

A pre-treated GCE surface was coated with a hydrophilic material consisting of electrochemically deposited polydopamine, which was followed by the deposition of tannic acid assisted by Fe^3+^ ions [66]. In the next step, the branched structure of four-armed polyethyleneglycol was grafted onto a modified interface via a layer-by-layer technique. To enhance *BRCA1* gene detection, AuNPs with thiol-modified oligonucleotides were finally deposited onto the modified surface (Figure 3). An impedimetric biosensor detected *BRCA1* with an LOD of 0.05 fM in a linear range from 0.1 fM to 10 pM [66] (Approach 1). A carbon paste electrode (CPE) modified with electrospun ribbon conductive nanofibers of polyethersulfone and nanotubes were employed for *BRCA1* detection by Ehzari et al. [67]. DNA was detected with LOD of 2.4 pM with high selectivity, stability, reproducibility, and with a recovery index in the range from 101.5% to 105.2% [67] (Approach 1).

Graphene oxide (GO) was successfully applied as a promising nanomaterial with high surface area for the detection of the *BRCA1* gene. Kazerooni and Nassernejad developed a biosensor for detection of *BRCA1* with LOD of 2 pM by applying supramolecular ionic liquids grafted on nitrogen-doped graphene aerogel-modified GCEs by electrochemical reading [68] (Approach 1). The single-stranded DNA probe for the detection of *BRCA1* 5382 insC mutation was immobilized onto GCE electrochemically patterned with RGO and gold nanoparticles (AuNPs) [69]. The impedimetric biosensor was able to specifically recognize targets with LOD of 1.0 × 10^−20^ M [69] (Approach 1). RGO was also applied in combination with polypyrrole polymer by Shahrokhiana et al. for *BRCA1* detection [70]. A pyrrole-3-carboxylic acid monomer was electrochemically polymerized and applied for probe immobilization. *BRCA1* was determined with LOD of 3 fM in a linear range of 10 fM–0.1 µM by DPV and EIS [70] (Approach 1).

In addition to the development of a biosensor for the detection of the *BRCA1* gene, a DNA biosensor for the detection of the ERBB2c gene (producing HER2 protein) and CD24c was also prepared [71]. GCE was modified by GO, to which 4-aminothiophenol as a linker was covalently attached via amine coupling. The linker was in the subsequent step applied for the attachment of AuNPs. Then, a DNA capture probe was deposited on AuNPs via SAM formation. Then, DNA for the ERBB2c target was hybridized with a surface-confined capture DNA probe. Finally, the electrochemical signal was generated by hybridization with a conjugation DNA probe linked to horseradish peroxidase (HRP). The biosensor detected the ERBB2c gene down to 0.16 nM and CD24 down to 0.23 nM [71] (Approach 1).

The phosphatidylinositol-4,5-bisphosphate 3-kinase catalytic subunit alpha gene (*PIK3CA* gene) as a circulating tumor DNA was detected by a biosensor employing a nanocomposite of MoS_2_ and poly(indole-6-carboxylic acid) as a surface-confined mediator, which was also applied for the covalent immobilization of –NH_2_-modified ssDNA [72]. First, the surface of CPE was modified with exfoliated MoS_2_ nanosheets, and then, it was incubated with a mediator monomer via π–π stacking with a subsequent potentiostatic polymerization of the monomer. Afterwards, ssDNA probes were covalently immobilized to such a modified electrode. The DNA biosensor could detect analytes down to 15 aM [72] (Approach 1).

### 4.2. Detection of MicroRNAs (MiRNAs)

The miRNAs are biomolecules consisting of 18–24 nucleotides that have a key role in biological processes such as cell proliferation, apoptosis, and tumorigenesis [73,74,75]. Abnormal expression has been observed in BC as well as in other cancer types [74,76].

GO was exploited as an effective part of several biosensors for miRNA detection. For example, the electrochemical nanobiosensor based on GCE that was step-by-step modified with GO and gold nanorods was fabricated for the detection of a serum miR-199a-5p level [77]. A thiolated oligonucleotide probe was immobilized on the modified electrode, and unspecific bindings were blocked by incubation with 6-mercapto-1-hexanol solution. The nanobiosensor exhibited LOD of 4.5 fM, which is a standard deviation of 2.9% for miR-199a-5p detection and a linear range from 15 fM to 148 pM [77] (Approach 1).

An impedimetric biosensor based on ZrO_2_–RGO nanohybrids-modified GCE coupled with a catalytic hairpin assembly signal amplification strategy determined miRNA-21 in the range from 10 fM to 100 pM with LOD of 4.3 fM [78]. H1 modified with –NH_2_ was covalently attached onto the ZrO_2_–RGO-modified GCE surface via poly(acrylic acid) using amine coupling chemistry. In the absence of the analyte (miRNA-21), H1 and H2 did not hybridize. When the analyte was present, the hairpin of H2 hybridized with the analyte, which caused opening of the closed structure of H2. Subsequently, H1 hybridized with the unfolded H2. After this, target miRNA was released due to the DNA strand displacement reaction. At the end, H2 was attached to the electrode surface, and targeted miRNA started another cycle. This caused the amplification of the detected signal, since several H2 molecules per one analyte molecule were attached to the electrode surface (Figure 4) [78] (Approach 1).

The miRNA sensor using methylene blue as a redox mediator was fabricated by Rafiee-Pour et al. with a linear range from 0.1 to 500 pM with LOD of 84.3 fM [79]. In the experiment, the GCE electrode was modified with the dispersion of oxidized MWCNTs. Afterwards, 1.0 μM ss-DNA was immobilized, and half of the modified electrodes were incubated with target miRNA. The second half of the electrodes was used as a control. Non-hybridized miRNA was removed from the surface with saline sodium citrate. Both types of electrodes were immersed into 4.0 μM methylene blue, which was intercalated into a double-stranded helix, and DPV was applied to evaluate the change of the electrochemical signal [79] (Approach 1).

Kilic et al. detected miRNA from cell lysates by using graphene-modified disposable pencil graphite electrodes [80]. The electrode was modified by an inosine substituted anti miRNA-2 probe. The analyte was detected with LOD of 2.1 μg/mL (EIS) or 5.8 μg/mL (DPV) [80] (Approach 1).

An enzyme-free biosensor based on a sandwich-type hybridization of two DNA probes with target miRNA was developed by Zouari et al. [81]. Thiol chemistry ensured the immobilization of a thiolated capture DNA onto the electrodes modified by a hybrid nanomaterial of RGO and AuNPs. Ferrocene-capped AuNPs were modified with streptavidin and conjugated with a biotinylated signal probe containing signal DNA. An enzymeless biosensor was able to determine the synthetic target miRNA with LOD of 5 fM (a linear range between 10 fM and 2 pM). Moreover, the biosensor was able to determine the target miRNA directly in diluted serum from BC patients. A 3-fold higher level of miRNA-21 was detected in serum samples of BC patients compared to a control [81] (Approach 3).

### 4.3. Detection of Mucins

Nowadays, there are more than 20 known types of mucins. They are encoded by MUC genes and represent high molecular weight glycoproteins expressed on epithelial cells. Aberrantly glycosylated mucins are expressed in cancer cells and serve as oncogenic molecules [82].

Nawaz et al. applied diazonium salt chemistry to modify single-walled carbon nanotubes (SWCNTs) for a biosensor development [83]. The MUC1 aptamer was immobilized onto modified SPCE via amine coupling. A DNA aptamer-based biosensor detected MUC1 with LOD of 0.02 U/mL with a linear range up to 2 U/mL [83] (Approach 1).

The MUC1 biosensor was also developed using GCE modified with core–shell nanofibers, MWCNTs, and AuNPs that were covalently modified with the anti MUC1-binding aptamer for the detection of MUC1 [84]. The impedimetric device using a soluble redox probe was able to detect MUC1 with LOD of 2.7 nM with a linear range up to 115 nM [84] (Approach 1).

Mouffouk together with colleagues applied bioconjugated self-assembled pH-responsive polymeric micelles loaded with ferrocene (Fc) and antiMUC1 antibodies as a signal probe [85]. The biosensor was able to detect MUC1 in a sample containing about 10 cells/mL [85] (Approach 2).

Nowadays, a novel 2D nanomaterial MXene (Ti_3_C_2_) due to its excellent electrical conductivity and large specific surface area with a large number of potential attachment binding sites is used as a conductive support for the immobilization of aptamer probes [53]. Wang et al. modified an electrode surface with MXene for the development of a MUC1 biosensor [86]. The Fc-labeled complementary DNA was bound onto MXene nanosheets to form a signal probe to amplify an electrochemical signal. GCE was modified by the electrodeposition of AuNPs with the MUC1 aptamer attached to the modified electrode via Au–S bonds. The modified electrode was blocked using bovine serum albumin (BSA) in order to resist non-specific interactions (Figure 5). Then, a signal probe was attached to the modified electrode via hybridization between complementary DNA and a MUC1 aptamer. Upon the interaction of MUC1 with such an electrode, the signal probe was detached from the working electrode, resulting in a decrease of an electrochemical signal (a signal-off response). This competitive aptasensor detected MUC1 with LOD of 0.33 pM with a linear range up to 10 mM [86] (Approach 3).

CA15-3 (290–400 kDa) represents a soluble form of mucin 1 (MUC1): a transmembrane protein on the apical cellular surface. MUC-1 is a glycoprotein with three domains. The association between BC and elevated expression of CA15-3 has been experimentally confirmed [87].

Santos et al. used imprinting technology with a CA15-3 imprint within an electropolymerized layer of polypyrrole for CA15-3 detection [88]. Polypyrrole was deposited on a fluorine-doped tin oxide conductive glass support in the presence of the analyte. Then, the analyte was removed from the imprinted layer with ethanol, and the biomimetic material was then incorporated in a polyvinylchloride plasticized membrane acting as a potentiometric ionophore. The best results were obtained with electrodes covered by the imprinted polymer without any lipophilic additive with LOD of 1.07 U/mL and a linear response from 1.44 to 13.2 U/mL for CA15-3 [88] (Approach 1).

A CA15-3 immunosensor based on RGO and CuS NPs was fabricated using gold screen-printed electrode [89]. Firstly, anti CA15-3 antibodies were immobilized on the electrode. Once the analyte CA15-3 was bound to the surface of the electrode, the electrochemical response toward catechol was decreased. The sensor reached LOD of 0.3 U/mL, a sensitivity of 1.88 μA/(μM cm^2^), and a linear response from 1.0 to 150 U/mL [89] (Approach 1).

Nakhjavani et al. prepared a sandwich-type of electrochemical immunosensor for the detection of CA15-3 [87]. Bare GE was incubated with streptavidin for 12 h with the subsequent immobilization of biotinylated anti-CA15-3 monoclonal antibodies. A considerable signal enhancement was reached due to the enhanced density of HRP delivered via streptavidin-coated magnetic beads (MBs) conjugated with biotinylated HRP and anti-CA15-3 antibodies. CA15-3 was detected employing the immunosensor in an electrolyte containing 0.1 M PBS pH 7.0 with a hydroquinone (HQ) as a redox mediator in the presence of H_2_O_2_ by CV and EIS techniques with LOD of 15 × 10^−6^ U/mL (a linear range from 50 to 15 × 10^−6^ U/mL). The lowest value of an electron-transfer resistance (R_et_) at a bare electrode increased after the addition of streptavidin onto the surface, as well as after adding monoclonal antibodies and finally after CA15-3 addition. The R_et_ values decreased after the addition of a detection label, confirming attachment onto the electrode surface [87] (Approach 2).

The nanostructure-based immunosensor was developed by applying the non-covalent functionalization of GO with 1-pyrenecarboxylic acid as a modified electrode interface for the immobilization of a primary antibody (Ab_1_) against the analyte [90]. Pre-treated GE were modified with a SAM of cysteamine, and the remaining empty places on the electrode were blocked with 2-mercaptoethanol. These electrodes were covalently patterned by GO already functionalized with 1-pyrenecarboxylic acid via amine coupling. Then, such modified electrodes were immobilized with monoclonal anti-CA15-3 Ab_1_, blocked with BSA, incubated with CA15-3, and after immunoreaction took place, they were incubated with a signal probe (Figure 6). MWCNTs supporting a high density of ferritin molecules together with secondary antibody (Ab_2_) against the analyte applied as a signal probe for the determination of CA15-3. MWCNTs were treated by a mixture of strong inorganic acids for the formation of carboxylic groups, for nanotube shortening, and for removing metallic and carbonaceous impurities. After the activation of MWCNTs, nanotubes were covalently modified by polyclonal anti-CA15-3 Ab_2_ and ferritin. CA15-3 was detected through an enhanced bioelectrocatalytic reduction of H_2_O_2_ mediated by HQ at the immunosensor-offered LOD of 0.01 U/mL in human serum samples using DPV [90] (Approach 3).

### 4.4. Detection of Human Epidermal Growth Factor Receptor-2 (HER2)

HER2 (185 kDa) i.e., human epidermal growth factor receptor-2, belongs to a family of receptor tyrosine kinases [91]. HER2 in BC is characterized by its high expression of growth factor receptor-related genes (*ERBB2*, *EGFR*, and/or *FGFR4*) and cell cycle-related genes [85].

There are several publications describing the development of biosensor platforms using various forms of graphene to enhance the selectivity and specificity of such devices. The in situ growth of 1D molybdenum trioxide anchored onto the 2D RGO via one-pot low-temperature hydrothermal synthesis and further functionalized using 3-aminopropyltriethoxysilane was fabricated as a suitable nanohybrid platform for HER2 detection [92]. In the following step, the surface conjugation of the monoclonal anti-HER2 antibodies onto the modified electrode was performed via amine coupling chemistry (Figure 7). The LOD of this nanohybrid-based immunosensor was 0.001 ng/mL, with a linear response in a concentration range of 0.001–500 ng/mL [92] (Approach 1).

An HER2 biosensor was prepared by the modification of GCE by a thin layer of RGO and SWCNTs to which a densely packed layer of AuNPs was deposited [93] (Approach 1). In the final step, the aptamer against HER2 was attached to the modified electrode and changes in the impedance were applied for the detection of HER2 with LOD of 50 fg/mL (a linear range from 0.1 pg/mL to 1 ng/mL). The recovery index of HER2 detection, when spiked into serum samples, was close to 100%, and the results of assaying HER2 levels in serum samples obtained by the biosensor device were in an excellent agreement with the ELISA method [93].

Arkan developed an impedimetric immunosensor using a hybrid nanomaterial modified electrode by the deposition of AuNPs and MWCNTs glued to the electrode by ionic liquid [94]. AuNPs were electrodeposited onto an electrode already patterned by MWCNTs and ionic liquid. Such an electrode was then immersed in an ethanol solution of 1,6-hexanedithiol. Then, another layer of AuNPs was deposited to which anti-HER2 antibodies were covalently grafted via amine coupling. It was found out that the charge transfer resistance increased linearly with increasing concentrations of HER2 antigen. The biosensor could detect HER2 in the linear range from 10 ng/mL to 110 ng/mL with LOD of 7.4 ng/mL. The results indicated the ability of HER2 detection in serum samples of BC patients, and such assays were in an excellent agreement with the results obtained by a commercial HER2 kit [94] (Approach 1).

An electrochemical molecularly imprinted polymer-based sensor (Figure 8) was developed for the detection of an extracellular domain of HER2 [95]. The sensor was prepared on a screen-printed gold electrode (AuSPE), where a molecularly imprinted layer was electropolymerized from a solution consisting of phenol and the analyte using the CV technique. The device exhibited a linear range for analyte detection from 10 to 70 ng/mL and LOD of 1.6 ng/mL, when DPV was applied as an electrochemical detection technique [95] (Approach 1).

Freitas et al. developed several biosensor devices for the detection of HER2 [96,97,98,99]. The first one was developed on SPCE modified either by AuNPs or combination of AuNPs with MWCNTs [96]. Such a modified electrode was then modified by primary anti-HER2 antibodies. Then, the biosensor was incubated with an analyte, and in the subsequent step, it was incubated with biotinylated secondary anti-HER2 antibodies. The electrochemical signal was generated by a final incubation of the biosensor by streptavidin-modified alkaline phosphatase, which catalytically reduced silver ions in the presence of 3-indoxyl phosphate. Under optimal conditions, the biosensors could detect HER2 in a concentration window of 7.5–50 ng/mL with LOD of 0.16 ng/mL (MWCNTs with AuNPs) or 8.5 ng/mL (AuNPs). The total assay time was 140 min, and the biosensor was applied for the analysis of HER2 spiked into serum samples [96] (Approach 1).

Malecka with colleagues constructed a cellulase-linked sandwich assay based on magnetic beads for HER2 detection [100]. The principle behind detection is the formation of an insulating layer consisting of nitrocellulose film on spectroscopic graphite electrode. HER2 interacts with primary aptamer/antibody-modified magnetic beads with the subsequent formation of a sandwich configuration on MBs by secondary aptamers/antibodies conjugated to cellulose. Once MBs are incubated with the electrode surface, nitrocellulose film is digested with the formation of holes within the film, resulting in a decrease of electrode capacitance (Figure 9). The chronocoulometry was measured for the determination of an electric charge, which was proportional to HER2 in the concentration window of 10^−15^–10^−10^ M HER2 with LOD of 1 fM and with an overall assay time within 3 h. HER2 spiked into serum samples was detected with a recovery index of (109 ± 3)% [100] (Approach 1).

A DNA-based biosensor for the detection of HER2 was designed by the modification of GE with a DNA tetrahedron containing an aptamer against HER2 [101]. An electrochemical signal was generated by a signal probe consisting of gold nanorods with deposited PdNPs (5 nm), anti-HER2 aptamer, and HRP. Upon interaction of the modified electrode with HER2, a sandwich configuration was completed by a final incubation of the electrode with the signal probe. The biosensor could detect the analyte with LOD of 0.15 ng/mL and within a linear range from 10 to 200 ng/mL. Finally, the biosensor was applied for the analysis of HER2 spiked into serum samples [101] (Approach 2).

Lah and co-workers constructed a sandwich immunosensor for HER2 detection based on PbS quantum dots (QDs)-conjugated secondary anti-HER2 antibody as a signal probe [102]. Firstly, PbS QDs were synthesized, and anti-HER2 antibodies were attached to them. The application of QDs provided advantageous features such as a straightforward synthesis and well-defined electrochemical stripping signal of Pb(II) through acid dissolution. Primary anti-HER2 antibodies were immobilized onto pre-treated activated SPCE to capture the analyte. In the final incubation step, the signal probe formed a sandwich configuration. The biosensor could detect analyte down to 0.28 ng/mL with a linear calibration range up to 100 ng/mL. The biosensor was tested for the analysis of HER2 spiked into serum samples with a recovery index in the range from 91% to 104% [102] (Approach 2).

In the next work of Freitas et al., primary anti-HER2 antibodies were immobilized on MBs (Figure 10) [97] (Approach 2). The whole immunocomplex sandwich was formed directly in the solution phase, and then it was magnetically transferred to the electrode surface. The total assay time was 205 min with LOD down to 2.8 ng/mL. The biosensor was applied for the analysis of HER2 spiked into serum samples with a recovery index of 95%–99% [97].

The advanced approach was achieved employing the core/shell CdSe@ZnS QDs as an electroactive detection probe for HER2 biosensing, requiring a total time assay of 2 h [98]. The sandwich configuration was formed on the SPCE involving primary and secondary anti-HER2 antibodies. The biosensor required only 40 μL of a sample volume with an LOD down to 2.1 ng/mL. The biosensor was applied for the analysis of HER2 spiked into serum samples with a recovery index between 104% and 106% [98] (Approach 2).

In next paper, the authors combined magnetic beads and core/shell streptavidin-modified CdSe@ZnS QDs as an electroactive detection probe for the affinity-based detection of HER2 with LOD of 0.29 ng/mL (a linear range of 0.50–50 ng/mL) [99]. The device was applied for the detection of HER2 spiked into serum samples with a recovery index of 100%–108%, assay time of 90 min, and with a good agreement with the reference ELISA method, which took 285 min to complete [99] (Approach 2).

Hartati et al. used a bioconjugate prepared by the covalent immobilization of anti-HER2 antibodies onto cerium oxide NPs previously modified by 3-aminopropyl trimethoxysilane (APTES) and polyethylene glycol-α-maleimide-ω-NHS (PEG–NHS–maleimide) [103]. Then, such a bioconjugate was covalently attached to SPCE modified by AuNPs. The interaction of HER2 with the modified electrode was analyzed by CV with a decrease of the peak current in the presence of the analyte (a signal-off approach). The biosensor could detect HER2 with an LOD of 34.9 pg/mL. The biosensor was finally used for the analysis of HER2 spiked into serum samples with a recovery index close to 100% [103] (Approach 3).

### 4.5. Detection of Carcinoembryonic Antigen (CEA)

CEA (180–200 kDa) is a glycoprotein that participated in cell adhesion. Normally, it is expressed by normal fetal intestinal tissue, and after birth, its expression is inhibited. The serum level can be increased in non-malignant diseases such as inflammatory bowel disease and also in many types of human cancers, such as gastric cancer, breast cancer, ovarian cancer, lung cancer, pancreatic cancer, and colorectal cancer [104].

Wang et al. developed a label-free aptasensor based on an electrochemiluminescent (ECL) strategy with ZnS–CdS NP-decorated molybdenum disulfide (MoS_2_, a 2D nanomaterial [105]) nanocomposite for CEA detection [106]. The GCE was firstly modified with layered MoS_2_ as an electrode matrix, and then ZnS–CdS NPs were electrodeposited directly onto MoS_2_/GCE. In the next step, chitosan and glutaraldehyde covered the electrode for the immobilization of an anti-CEA aptamer. The aptasensor was completed by a final incubation with BSA to suppress non-specific interactions. The ECL aptasensor showed a linear range from 0.05 to 20 ng/mL with an LOD of 0.031 ng/mL. CEA spiked into human serum was analyzed with a recovery index in the range from 80% to 111%. The method was also applied for the determination of CEA in 8 human serum samples with an excellent agreement with a reference analytical method, showing the clinical application of the approach [106] (Approach 1).

Paimard with co-workers developed an immunosensor for CEA detection based on the CPE surface covered by the core–shell nanofibers prepared by electrospinning [107]. A nanofiber was made of honey (a core) electrospun with polyvinylalcohol (a shell) formed by a coaxial approach. Electrospun nanofibers were decorated with AuNPs and MWCNTs. Subsequently, anti-CEA antibodies were immobilized on the electrode surface. The impedimetric immunosensor exhibited high sensitivity toward the CEA biomarker with LOD of 0.09 ng/mL and with a linear range up to 125 ng/mL. The biosensor was applied for the analysis of CEA in human serum samples. Significantly higher levels of CEA were found in the serum samples of cancer patients compared to control, which was also verified using ELISA [107] (Approach 1).

Wang with colleagues employed flower-like Ag/MoS_2_/RGO nanocomposites deposited onto GCE for CEA label-free detection with LOD of 1.6 fg/mL through the electrocatalytic H_2_O_2_ reduction [108]. Firstly AgNPs and GO were synthesized by a seed-mediated Lee–Meisel method and by an improved Hummer’s method, respectively. Next, MoS_2_/RGO was synthesized by applying Na_2_MoO_4_•2H_2_O and thiourea to obtain the final Ag/MoS_2_/RGO nanocomposite. Anti-CEA antibodies were conjugated to the surface of AgNPs via amino groups for CEA determination in a wide concentration range from 0.01 pg/mL to 100 ng/mL. The analysis of CEA spiked into serum samples revealed a recovery index that was very close to 100% [108] (Approach 1). Another electrochemical platform for the detection of CEA using H_2_O_2_ reduction was developed by Su et al. [109]. Two-dimensional nanomaterial MoS_2_ was modified by Prussian blue NPs, and such a hybrid nanomaterial was then deposited on GCE. The biosensor was finalized by the covalent immobilization of anti-CEA antibodies with subsequent surface blocking by BSA. CEA determination through the non-enzymatic detection of H_2_O_2_ offered LOD of 0.54 pg/mL (a linear range from 0.005 to 10 ng/mL). When the biosensor was applied for the detection of CEA spiked into human serum samples, a recovery index from 95% to 102% was obtained [109] (Approach 1).

Another sandwich-type electrochemical immunosensor for the determination of CEA was based on SPCE modified by AgNPs and RGO to which primary anti-CEA antibodies were immobilized [110]. After the electrode interface was incubated with an analyte, secondary anti-CEA antibodies labeled with HRP were added to complete a sandwich configuration, and a reduction of H_2_O_2_ was detected electrochemically. The modified SPCE-based biosensor detected CEA with LOD down to 0.035 µg/mL (a linear range of 0.05–0.50 µg/mL) [110] (Approach 1).

Rizwan et al. applied a layer-by-layer deposition of AuNPs, carbon nano-onions, SWCNTs, and chitosan layers onto GCE for the construction of a CEA immunosensor [111]. SWV was applied as an output signal in the presence of a soluble redox probe, and the device offered a linear range from 100 fg/mL to 400 ng/mL with LOD of 100 fg/mL for the detection of CEA. Only one serum sample spiked with three different CEA concentrations was applied for a clinical evaluation of the biosensor with recovery index in the range of 105%–110% [111] (Approach 1).

Wang and Hui [112] utilized the zwitterionic poly (carboxybetaine methacrylate) as a superhydrophilic matrix for the immobilization of anti-CEA antibodies and also as a layer resisting non-specific interactions. GCE was modified via electrodeposition by polyaniline nanowires, which were then activated to covalently graft zwitterionic monomers to the interfacial layer. In the subsequent step, a polymeric form of the zwitterions was prepared using UV irradiation. Finally, anti-CEA antibodies were immobilized via amine coupling. CEA concentration in the range from 1.0 × 10^−14^ g/mL to 1.0 × 10^−10^ g/mL with LOD of 3.05 fg/mL was determined by a DPV method. Four serum samples were analyzed by the biosensor, with the CEA values obtained being in excellent agreement with the reference ECL method, and when CEA was spiked in serum samples, a recovery index between 94% and 104% was obtained [112] (Approach 1).

Kumar et al. [113] functionalized ultrathin 2D nanomaterial Ti_3_C_2_ MXene nanosheets with aminosilane for the covalent immobilization of anti-CEA antibodies for ultrasensitive CEA detection with LOD down to 18 fg/mL (Figure 11). The label-free biosensor exhibited a linear detection range of 0.0001–2000 ng/mL using a soluble redox probe [Ru(NH_3_)_6_]^3+^. The biosensor was applied for CEA detection when spiked into a human serum sample with a recovery index from 99% to 101% [113] (Approach 1).

Yang and co-workers utilized a label-free amplification strategy based on an Au-Ag/RGO nanohybrid prepared using dopamine as a reducing agent, which was deposited on GCE [114]. Such a modified electrode was then used for the immobilization of anti-CEA antibodies. CEA was detected by the decrease of an electrochemical signal due to the oxidation of AgNPs present on a signal probe upon incubation with an analyte with LOD of 0.286 pg/mL (a linear range from 0.001 ng/mL to 80 ng/mL). A serum sample spiked with 3 different CEA concentrations was successfully analyzed by the biosensor with a recovery index of 96%–107% with excellent agreement with an ELISA method [114] (Approach 1).

Gu et al. [115] integrated ferrocene (Fc) derivative and AuNPs into their biosensor for CEA detection in order to increase the conductivity of the sensing surfaces and increase ferrocene loading. Firstly, AuNPs were reduced from chloroauric acid with trisodium citrate as a reducing agent, and subsequently, polyclonal secondary anti-CEA antibodies were immobilized onto their surface via physisorption. Further chemisorption of the electroactive ferrocene molecules in a form of thiolated ferrocene chains was accomplished on AuNPs. Finally, PEG8000 was applied to stabilize AuNPs, and repeated centrifugation was applied to remove excess antibodies and Fc, and such a bioconjugate was applied as a signal probe. Pre-treated GEs were first modified with lipoic acid *N*-hydroxysuccinimide ester to attach primary antibodies, and the surface was blocked with ethanolamine. After CEA was affinity captured on the modified electrode, the sandwich configuration was completed by incubation with a signal probe. The developed biosensor exhibited LOD of approximately 0.01 ng/mL (a linear range up to 20 ng/mL), when detecting CEA using a SWV method with a good performance after storage for 3 weeks (91.8% of the original response) [115] (Approach 2).

Wei et al. developed an electrochemical ratiometric method for CEA detection [116]. The method was based on an AuNPs functionalized Cu_2_S-CuS/graphene composite as a SPCE-modifying nanomaterial to which primary anti-CEA antibodies were immobilized. A signal probe was developed using CeO_2_ NPs modified by deposited AuNPs to which secondary anti-CEA antibodies and toluidine blue (TB) as a redox mediator were covalently immobilized. The adsorption capacity toward toluidine blue was improved with carboxymethyl chitosan (CMC)-doped ionic liquids containing active groups such as −OH, −COOH, and –NH_2_. The change of dual signals “Δ*I* = Δ*I*_TB_ + |Δ*I*_Cu2S-CuS_|” (Δ*I*_TB_ and |Δ*I*_Cu2S-CuS_| present the change values of the oxidation peak currents of toluidine blue and Cu_2_S-CuS, respectively) was applied as the response signal for the quantitative determination of CEA with LOD of 0.78 pg/mL (a linear range of 0.001–100 ng/mL) (Figure 12). The biosensor was applied for the analysis of CEA in one serum sample, and the CEA level found out by the biosensor device was in an excellent agreement with an ELISA method [116] (Approach 3).

Another type of a label-based sandwich-type electrochemical immunosensor for CEA determination was developed by Li et al. [117], who used amino functionalized magnetic graphene sheets loaded with Au@Ag core–shell NPs to adsorb Ni^2+^ and secondary anti-CEA antibodies as a signal probe to reduce H_2_O_2_. AuNPs electrodeposited from HAuCl_4_ solution onto GCE improved the immobilization of primary anti-CEA antibodies and the device exhibited an LOD of 0.07 pg/mL (a linear range from 0.1 pg/mL to 100 ng/mL). The biosensor offered a recovery index close to 100% for the determination of CEA spiked into serum samples [117] (Approach 3).

A similar strategy based on the application of multiple types of nanoparticles for the detection of CEA was also applied by Wu et al. [118]. GCE was patterned by aminated-graphene sheets to which primary anti-CEA antibodies were covalently immobilized using glutaraldehyde. A signal probe was made of magnetic NPs covered by a shell made of a MnO_2_ layer with a deposition of PtNPs to which secondary anti-CEA antibodies were immobilized. CEA was detected with an LOD of 0.16 pg/mL in a linear range from 0.5 pg/mL to 20 ng/mL. Serum samples spiked with different CEA concentration provided reliable results with a recovery index from 95% to 106%, and the assay was validated using an ELISA method [118] (Approach 3).

### 4.6. Dual-Target Analysis

The dual-target detection of miRNA-21 and MUC1 based on a dual catalytic hairpin assembly was performed by Li and co-workers [119]. GCE was modified by Au nanoflowers to which hybridization probe 1 was immobilized to recognize miRNA-21. After incubation with miRNA-21, the electrode was incubated with a hybridization probe 2 conjugated with QDs, resulting in an increase of ECL signal (Cycle I, Figure 13). When such an electrode was incubated with anti-MUC1 aptamer and MUC1, both molecules were attached to the modified electrode surface. Incubation with a hybridization probe 3 conjugated with AuNPs in the subsequent step replaced the anti-MUC1 aptamer from the electrode surface and due to a fluorescence resonance energy transfer between QDs and AuNPs, a decrease of ECL signal was observed (Cycle II, Figure 13). The biosensor detected miRNA-21 with LOD of 11 aM and MUC1 with LOD of 0.40 fg/mL. When both analytes were spiked into human serum samples, a recovery index between 98% and 103% was obtained [119] (Approach 3).

### 4.7. Detection of Other Potential BC Biomarkers

In the next part, we will focus on an electrochemical performance for the detection of less known biomarkers present in the serum of BC patients.

Cancer antigen 27.29 (CA27.29, 250–1000 kDa) is a soluble form of glycoprotein MUC1. It is expressed mainly in BC, but CA 27.29 levels can also be elevated by colon, stomach, kidney, lung, ovary, pancreas, and liver cancers as well as other non-cancerous conditions such as benign breast disease, kidney, and liver diseases [120]. Alarfaj et al. constructed a label-free electrochemical immunosensor based on an Au/MoS_2_/RGO nanocomposite system [121]. First, a hybrid Au/MoS_2_/RGO nanocomposite was deposited on the GCE surface. Then, anti-CA 27-29 antibodies were immobilized on the modified electrode surface for selective capture of the analyte via affinity interactions. A signal amplification strategy was achieved by a synergy of all nanomaterial components of the nanocomposite to reduce H_2_O_2_. The biosensor could detect analyte down to an LOD of 0.08 U/mL. The device was finally applied for analysis of the analyte in 25 human serum samples with an excellent agreement with an ELISA method, and when CA27.29 was spiked into serum samples, an excellent recovery index of 96%–100% was obtained [121] (Approach 1).

Urokinase-type plasminogen activator receptor (uPa) belongs to cell membrane receptors with their expression increased in a number of different types of human cancers, including BC [122]. An immunosensor based on fluorine-doped tin oxide was modified with graphene nanosheets to enhance the loading of covalently immobilized antibodies [122]. The immunosensor could detect the analyte down to 4.8 fM using DPV assays in the presence of a soluble redox probe. The device offered a good stability (75% of an initial activity observed after 4 weeks) with the ability to detect an analyte spiked into serum samples [122] (Approach 1).

Tissue plasminogen activator (tPa, 20–45 kDa) belongs to serine proteases (enzymes ensuring cleaving peptide bonds in proteins). As a result of this fact, the protein is essential in the human body in relation to angiogenesis in cancer cells [123]. The protein was detected with LOD of 0.026 ng/mL in a linear range from 0.1 to 1.0 ng/mL [124]. A label-free biosensor was fabricated by the functionalization of SWCNTs with antibodies immobilized, and such a bionanoconjugate was subsequently immobilized onto a GCE surface (Figure 14) [124] (Approach 1).

### 4.8. Detection of BC Cells

Circulatory tumor cells (CTC) are released from tumors and circulate in the bloodstream at a low concentration of up to 10 cells/mL, while the whole blood contains 10^9^ erythrocytes and 10^6^ leucocytes/mL [30]. This is why the detection of CTC is quite challenging.

The detection of CTCs, which are present in the blood at a very low level, is highly challenging and has not been done using affinity-based approaches. In the following text, we discuss some detection principles for the analysis of BC cells with some approaches potentially applicable for the analysis of CTCs. More details about the electrochemical detection of BC cells can be found elsewhere [125].

The Michigan Cancer Foundation-7 (MCF7) cell line is the most frequently studied BC cell line [126], since it is a suitable model for studying the development/progression of BC and anticancer drug therapies. The cells are non-invasive, expressing estrogen as well as progesterone receptors [127].

An interesting method for the electrochemical detection of CTCs within a microfluidic channel was proposed by Gurudatt et al. (Figure 15) [128]. Cells differing in their size, surface charge, and chemical state on the cell surface were effectively separated using such a device. In order to detect CTCs in an effective way, the surface of channels was chemically modified with an electrochemical polymerization of a monomer. In the subsequent step, a lipid layer by the deposition of phosphatidylserine was formed on the surface of the channels. In order to electrochemically detect cells, such cells were loaded with daunomycin prior to separation. Three different types of cancer cell lines were used for optimization of the assay, and optimized assay conditions allowed detecting single cells (approximately 7 cells/mL). Finally, the device was applied for the detection of CTCs from 37 cancer patients with (92.0 ± 0.5)% efficiency. The results showed differences in the retention time for different types of CTCs produced by different cancer types, suggesting differences in the size, surface charge, and chemical state on the cellular surface [128]. Another microfluidic electrochemical approach for the detection of CTCs was based on the measurement of changes in the impedance of the polydimethylsiloxane-based channel on a glass slide during the passage of CTCs [129]. A narrow constriction-based sensor was designed in a way allowing the passage of red and white blood cells without any restrictions, while much larger tumor cells needed to squeeze/deform in order to pass through the channel, causing changes in the impedance of the channel. As a result, only cancerous cells were able to generate an electrochemical signal in a label-free format, while smaller blood cells did not generate any measurable signal (Figure 16). The device was tested by an analysis of murine blood spiked with prostate or breast cancer cells with a throughput of 1 μL *per* min, but the throughput can be increased by analysis run in parallel. A signal processing of data generated was done automatically using MATLAB. The authors claim that false positive results can be obtained due to the presence of non-blood or non-cancer cells in blood such as epithelial cells, and this why the pre-enrichment of CTCs was suggested [129].

Anti-MUC1 aptamers, hybrid AuNPs, and carbon dots (Au@CDs) modifying GE were applied for the label-free ECL detection of circulating MCF-7 cells (MCF-7 CTCs) [130]. The biosensor detected MCF-7 CTCs down to 34 cells/mL with a linear range up to 10,000 cells/mL. MCF-7 cells spiked into serum samples were in addition clinically tested with an obtained recovery index of 93–117% [130] (Approach 1). Tian et al. [131] investigated MCF-7 CTCs by using a supporting RGO/AuNPs composite deposited on GCE with a catalytic CuO nanozyme used as a signal probe (Approach 3). MCF-7 CTCs membranes contain specific a MUC1 protein, which was recognized by the MUC-1 aptamer. The reached LOD was as low as 27 cells/mL (a linear range from 50 to 7 × 10^3^ cells/mL). MCF-7 cells were further successfully studied and determined by applying aptamer-based electrochemical biosensors [132]. A DNA aptamer was immobilized onto AuNPs supported by α-cyclodextrin on GE (Approach 1). The aptasensor determined MCF-7 cells in the range of 328–593 cells/mL with LOD of 328 cells/mL, when cells were lysed and an intracellular level of platelet-derived growth factor was electrochemically determined [132]. Yang with colleagues [133] utilized GCE modified by several nanomaterials using a layer-by-layer deposition process incorporating 3D graphene, Au nanocages, and MWCNTs to which primary antibodies were immobilized (Approach 1). Once the biosensor was incubated with MCF-7 cells, a sandwich configuration was established by incubation with secondary antibodies linked to DNA. In the next step, complementary DNA was applied, and to double-stranded DNA, methylene blue as a redox mediator was intercalated and detected using SWV. The biosensor detected BC cells with LOD of 80 cells/mL (a linear range of 1.0 × 10^2^–1.0 × 10^6^ cells/mL) and exhibited satisfactory stability [133]. Wang et al. [134] fabricated a sensitive sandwich-based aptamer biosensor for the label-free electrochemical detection of cells (Approach 2). The sensor was based on GE modified with polyadenine (polydA)-aptamer recognizing MUC1 on the surface of cells. A signal probe was designed by the immobilization of an aptamer recognizing MUC1 protein on an AuNP/GO hybrid nanomaterial. MCF-7 cells were recognized by polydA-aptamer, and then, a sandwich configuration was completed by incubation with a signal probe. BC cells were detected via a DPV method with LOD of 8 cells/mL and a linear range from 10 to 10^5^ cells/mL using a soluble redox probe with satisfactory selectivity [134]. An interesting approach in order to differentiate between different BC cell lines was based on the detection of H_2_O_2_ produced by the cells [135]. A sandwich consisting of synthesized Bi_2_Se_3_ NPs as 3D topological insulators between the gold electrode and another Au-deposited thin layer was designed by Mohammadniaei et al. as a nanostructured working electrode. In order to detect H_2_O_2_ in an ultrasensitive fashion, the immobilization of double-stranded DNA loaded with Ag^+^ ions was established through the Au–thiol interaction of thiolated DNA (Figure 17) (Approach 1). The developed biosensor showed LOD of 10 × 10^−9^ M for H_2_O_2_ with a dynamic range from 0.10 × 10^−6^ M to 27.3 × 10^−6^ M and a short response time of 1.6 s. The biosensor could distinguish the MCF-7 cell line from the MDA-MB-231 cell line based on the H_2_O_2_ produced [135].

In the next study, a H_2_O_2_ sensor employing a trimetallic AuPtPd nanocomposite and RGO nanosheets deposited on GCE was applied for the electrocatalytic detection of H_2_O_2_ reduction with LOD of 2 nM (a linear range from 0.005 μM to 6.5 mM) [136]. The biosensor was applied for the detection of H_2_O_2_ released by two BC cell lines (MDA-MB-231 and T47D) [137] (Approach 1).

Luo et al. applied hexagonal carbon nitride tubes as a photoactive material to determine the photocurrent in the presence of MCF-7 cells (Approach 1) [137]. The cells were detected down to 17 cells/mL (a linear range from 100 to 1 × 10^5^ cells/mL). Glutaraldehyde was utilized as a cross-linker for the covalent immobilization of anti-MUC1 aptamers for the affinity capture of cells via surface-expressed MUC1. A clinical applicability of the biosensor was proved by the detection of cells spiked into blood samples at three different concentrations with a recovery index of 96%–104% [137].

Safavipour et al. developed an aptasensor using a hybrid nanomaterial composed of TiO_2_ nanotubes attached to GO via UV irradiation [138]. GCE was modified by such a hybrid nanomaterial with the subsequent immobilization of anti-MUC1 aptamers for the affinity capture of MCF-7 cells via surface-expressed MUC1 proteins. An EIS-based device was able to ultrasensitively detect MCF-7 cells with LOD of 40 cells/mL within a linear concentration range from 10^3^ to 10^7^ cells/mL [138] (Approach 1).

GE modified by non-spherical AuNPs was made by electrodeposition in the presence of a shape-controlling agent for achieving an increased electrode active area [139]. A thiolated aptamer recognizing BC cells MDA-MB-231 was chemisorbed on the modified electrode surface, and the cells were electrochemically detected down to 2 cells/mL in an electrolyte using a soluble redox probe. The device was also applied for the analysis of cells spiked into blood serum samples with LOD of 5 cells/mL [139] (Approach 1).

Two immunomagnetic biosensors, which were described in Section 4.4. “Detection of HER2”, were also applied for the detection of BC cells [97,99]. The first biosensor was applied for the determination of two BC cell lines: HER2^+^ SK-BR-3 and HER2^−^ MDA-MB-231 via surface-expressed HER2 proteins using Ag ions and 3-indoxyl phosphate for a signal generation [97] (Approach 2). The biosensor could detect cells in the linear range of 100–10,000 cells/mL with an LOD of 3 cells/mL [97]. Another immunomagnetic biosensor was applied for detection of the same BC cell lines as the first one [99] and an additional MCF-7 (a cell line with low HER2 expression) cell line via surface-expressed HER2 with a signal generated by a stripping voltammetry of Cd ions released from QDs (Approach 2). The selectivity toward SK-BR-3 cells was confirmed. A concentration-dependent signal that was 12.5× higher than the signal obtained for the HER2-negative cells (MDA-MB-231) and LOD of 2 cells/mL was obtained [99].

Cancer stem cells were discovered by Al-Hajj in 2003 [140]. Cancer stem cells were detected using a nanobiosensor with a thiolated aptamer against the CD44 surface protein immobilized on GE via chemisorption [141]. After stem-like cells were captured on the electrode surface, the electrode was modified by a self-assembled peptide-based multifunctional nanofiber containing CD44 binding protein and –N_3_ groups, which were subsequently used for the clicking of AgNPs applied as a redox probe for an electrochemical signal generation. The LOD of the device was 6 cells/mL, and the device offered a wide linear range (from 10 cells/mL to 5 × 10^5^ cells/mL). The selectivity of the device was successfully proved by the analysis of three different cancerous cell lines [141] (Approach 2).

### 4.9. Detection of Exosomes and Exosomal Content

Exosomes are characterized as endosome-derived vesicles involving the signal transduction within intercellular communication and in extracellular matrix remodeling. Exosomes are membrane-bound particles with a lipid bilayer structure carrying precious cargo: biomolecules that could be used as cancer biomarkers for more accurate cancer diagnostics in the future [51]. An increased number of exosomes circulating in body fluids was observed for cancer patients compared to healthy individuals, and a change in the exosome level can be applied as a diagnostic cancer biomarker on its own [51].

Kilic et al. fabricated a label-free electrochemical sensor to measure the increased release of nanoscale extracellular vesicles from the BC cell line, MCF-7, due to CoCl_2_-induced hypoxia [142]. A pre-treated surface of AuSPE was modified with 11-mercaptoundenoic acid with the subsequent activation of –COOH groups for the attachment of neutravidin, which was applied for the immobilization of biotinylated anti-CD81 antibodies on the surface (Figure 18). Such a sensor was able to detect extracellular vesicles with LOD of 77 particles/mL or 379 particles/mL using EIS and DPV, respectively [142].

Exosomes released from 4 BC cell lines were detected using a magneto-mediated electrochemical sensor [143]. Magnetic beads were modified with anti-CD63 aptamer for the capture of exosomes. The selective detection of 4 proteins (MUC1, HER2, EpCAM, and CEA) on the exosomal surface was achieved by using silica NPs modified with respective aptamers. Silica NPs were also functionalized using mercapto–ferrocene derivative. The sandwich structure magnetic beads–exosomes–SiNPs was separated using a magnet, and the ferrocene derivatives were released from the sandwich using dithiothreitol. Ferrocene derivatives released from the sandwich were electrochemically detected on SPCE modified by a GO layer. Using this approach, 4 different biomarkers on 4 different cell lines were sensitively detected (Figure 19) (Approach 2). The sensor was clinically tested by the analysis of expression profile of 4 proteins on exosomes isolated from one BC patient and one healthy individual, confirming statistically higher levels of all 4 exosomal proteins when using BC serum compared to the serum of a healthy individual [143].

Moura et al. prepared an electrochemical immunosensor for the detection of exosomes derived from three cell lines (MCF7, MDA-MB-231, and SKBR3) [144]. Exosomes were captured to MPs, which were modified with antibodies against general tetraspanins CD9, CD63, and CD81, as well as against specific receptors of cancer (CD24, CD44, CD54, CD326, and CD340) (Figure 20). Exosomes were immobilized on magnetic particles (MPs) in a direct and in an indirect format. The direct format was based on incubation of the exosomes–MP with the antiCD63-HRP antibodies with a final electrochemical signal readout. The indirect format was based on incubation of the exosomes–MPs with antiCDX mouse monoclonal antibodies (CDX being either CD9, CD24, CD44, CD54, CD63, CD81, CD326, or CD340 biomarkers) and the indirect labeling with antimouse–HRP antibodies. Hydroquinone was used as a mediator. The study also found out that there are differences in the size and amount of exosomes depending on the exosome origin. Moreover, the level and size distribution of exosomes purified from healthy individuals was strikingly different from the exosomes purified from BC patients. The approach offered LOD as low as 81 exosomes/μL. The method could be applied to distinguish exosomes from healthy donors and those isolated from BC patients [144] (Approach 2).

Luo et al. developed a ratiometric electrochemical DNA biosensor employing an immobilized locked nucleic acid (LNA)-modified in a form of a “Y” shape-like structure for the detection of miR-21 present in exosomes released by MCF-7 cell lines [145] (Approach 1). GCE was modified by a polylysine film to which a DNA probe 1 labeled with methylene blue was covalently attached followed by hybridization with a DNA probe 2 labeled with ferrocene. The DNA probe 2 was attached to the electrode in a way such that the ferrocene redox moiety was in proximity to the electrode surface, while methylene blue redox moiety was exposed to the solution phase (Figure 21). Upon binding of the analyte miRNA-21, a DNA probe 2 labeled with ferrocene was released from the electrode surface, leaving behind only a DNA probe 1 with a surface-confined methylene blue redox moiety. Thus, upon analyte binding, an increase (“signal-on” response) in the DPV signal for methylene blue was observed with a decrease (“signal-off” response) of the DPV peak attributed to ferrocene. Both single signal responses (i.e., “signal-on” response and “signal-off” response) exhibited significant signal variation, while a ratiometric signal was highly stable. When the biosensor response was expressed in a form of a ratiometric signal, the device offered LOD of 2.3 fM with a linear range from 10 to 70 fM. The biosensor exhibited high specificity with a negligible response obtained for single- and double-mismatched RNA sequences. The device showed high assay accuracy for the analysis of miRNA-21 released from exosomes produced by a MCF-7 cell line, which was confirmed by a reference analytical method [145].

## 5. Conclusions and Perspectives

This review provides evidence that electrochemical detection principles can offer ultrasensitive detection platforms for the detection of various BC biomarkers with LODs in some cases down to the single molecule level thanks to the use of a wide range of nanoparticles (Table 2). Such biosensors discussed in this review were mainly based on modification of the electrodes by nanomaterials (i.e., Approach 1), followed by the design of signal probes based on nanomaterials (i.e., Approach 2) with the design of a hybrid approach using nanomaterials for the modification of electrodes and for the design of signal probes (i.e., Approach 3). In a significant number of studies, the clinical performance of the nanobiosensors was validated by using human serum samples. Unfortunately, only a minor fraction of papers was focused on the detection of true concentration of BC biomarkers in human serum, but rather analyte spiking into serum samples was applied to validate the clinical usefulness of the nanobiosensors developed. Furthermore, only a limited number of papers dealt with the validation of nanobiosensing by a standard reference method i.e., ELISA. Such comparison is really needed to verify the reliability of nanobiosensors for the analysis of cancer biomarkers in complex samples such as serum samples from BC patients.

In order to really verify the clinical usefulness of the nanobiosensing approach, a larger number of human serum samples divided into two cohorts—BC patients and healthy (non-cancerous) individuals—need to be analyzed for the level of BC biomarkers in order to see if a particular biomarker is present in serum samples of BC patients at a statistically higher level compared to serum samples of healthy individuals. At the same time, such a comparison needs to be evaluated in the form of an ROC (Receiver Operating Curve) with an AUC (Area Under Curve) determined, and only such information can be then applied for the direct comparison of a clinical performance of nanobiosensing with standard immunoassays. The other aspect worth investigating in the future is a multiplexed format of analysis, when several samples are run in a parallel or when several biomarkers are detected in a single sample in parallel. So far, there is only one study describing the simultaneous analysis of miRNA-21 and MUC1, and only one study that determined the level of 4 different proteins on an exosomal surface. Although the electrochemical sensing approach is an ideal tool for integration into lab-on-a-chip platforms, we have not identified any single study analyzing BC biosensors in such an advanced assay platform using electrochemical sensing.

It is also of the utmost importance to deal with the non-specific binding of proteins from complex samples especially in cases when an electrochemical signal readout is not done in a sandwich configuration and rather detects changes in the interfacial properties of the interfacial layer after incubation with a sample i.e., impedimetric signal reading.

## Figures and Tables

**Figure 1 sensors-20-04022-f001:**
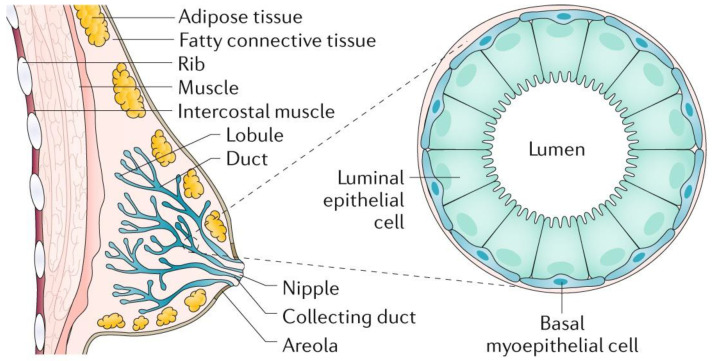
All breast cancers (BC) arise in the terminal duct lobular units (the functional unit of the breast) of the collecting duct. The histological and molecular characteristics have important implications for therapy. Several classifications based on molecular and histological characteristics have been developed. Reprinted by permission from Nature, Ref. [26], Copyright 2019.

**Figure 2 sensors-20-04022-f002:**
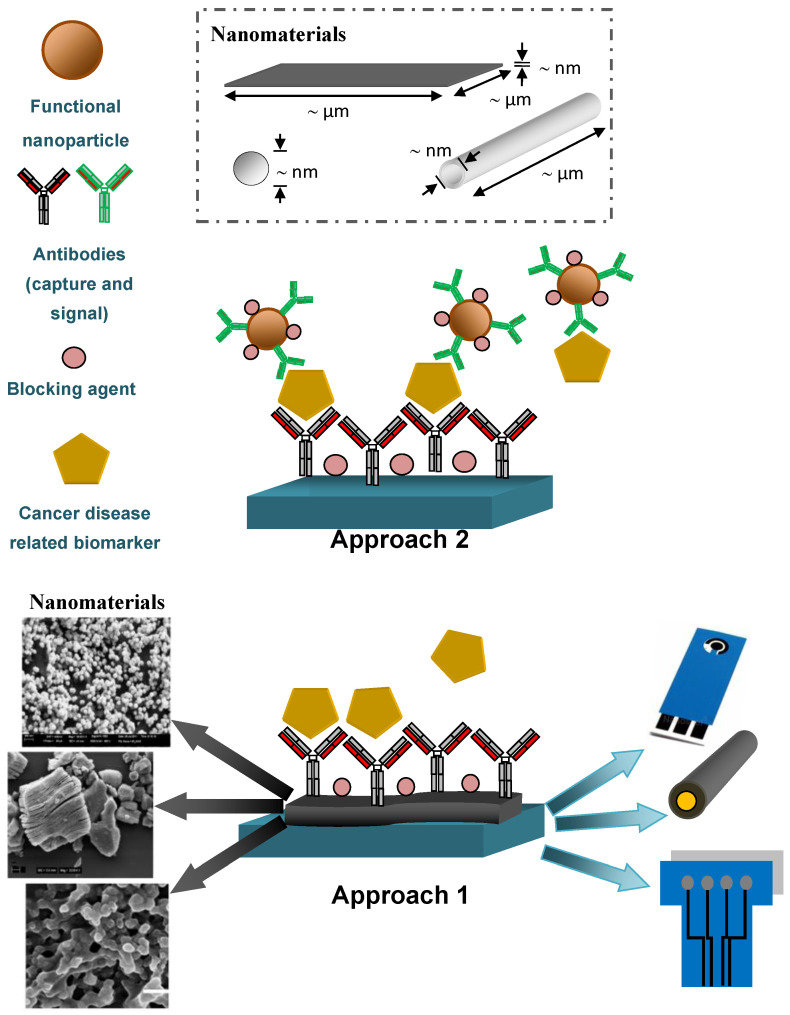
Schematic illustration of two different approaches applicable for the enhanced biosensing of cancer biomarkers using functional nanomaterials/nanoparticles either to enhance the electrode area, accessibility of analytes toward the interface, or interfacial properties with capture biorecognition elements (antibodies) immobilized (*Approach 1*) or for enhanced signal generation using a signal probe with biorecognition elements (antibodies) immobilized on the electrode (without being modified by nanoparticles) of a signal probe (*Approach 2*). Please note that we recognize *Approach 3* (not shown in the figure) applied to design biosensor devices by a combination of the nanomaterial/nanoparticle-modified electrode (*Approach 1***)** with the use of a signal nanoprobe (*Approach 2***)** within one biosensor device.

**Figure 3 sensors-20-04022-f003:**
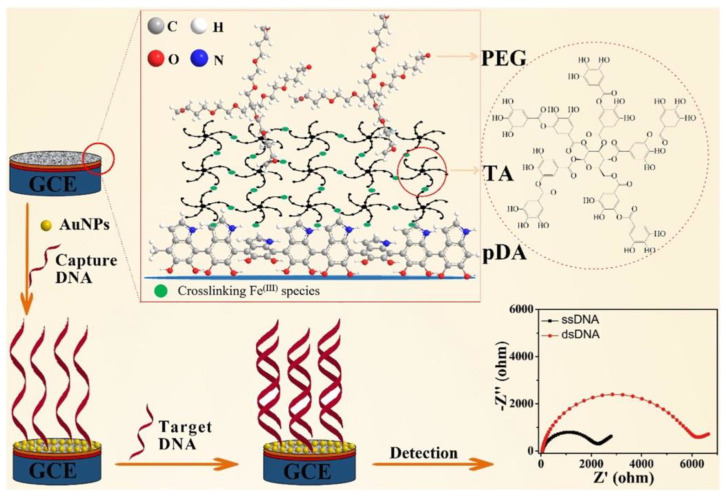
Fabrication process to design nanobiosensors for the detection of *BRCA1*. Reprinted from Ref. [66], Copyright (2017), with permission from Elsevier.

**Figure 4 sensors-20-04022-f004:**
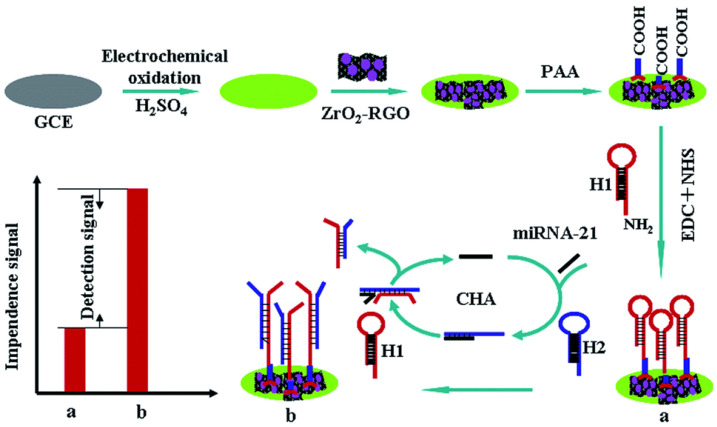
Immobilization of the capture probe H1 with subsequent miRNA-21 detection. For more information, please see the text. Published with permission by The Royal Society of Chemistry, Ref. [78].

**Figure 5 sensors-20-04022-f005:**
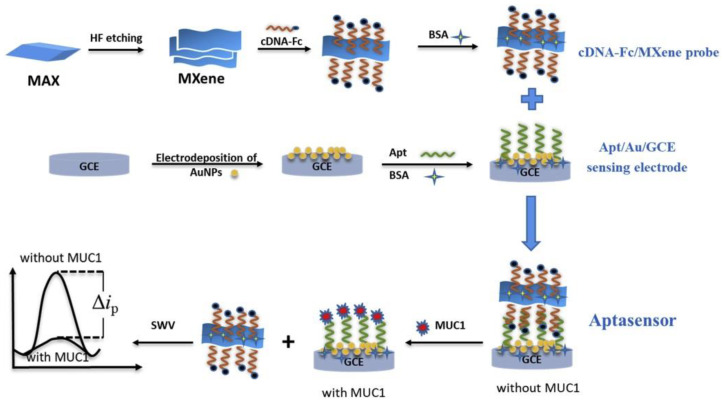
Procedure for fabrication of the competitive electrochemical aptasensor. Reprinted from Ref. [86], Copyright (2020), with permission from Elsevier.

**Figure 6 sensors-20-04022-f006:**
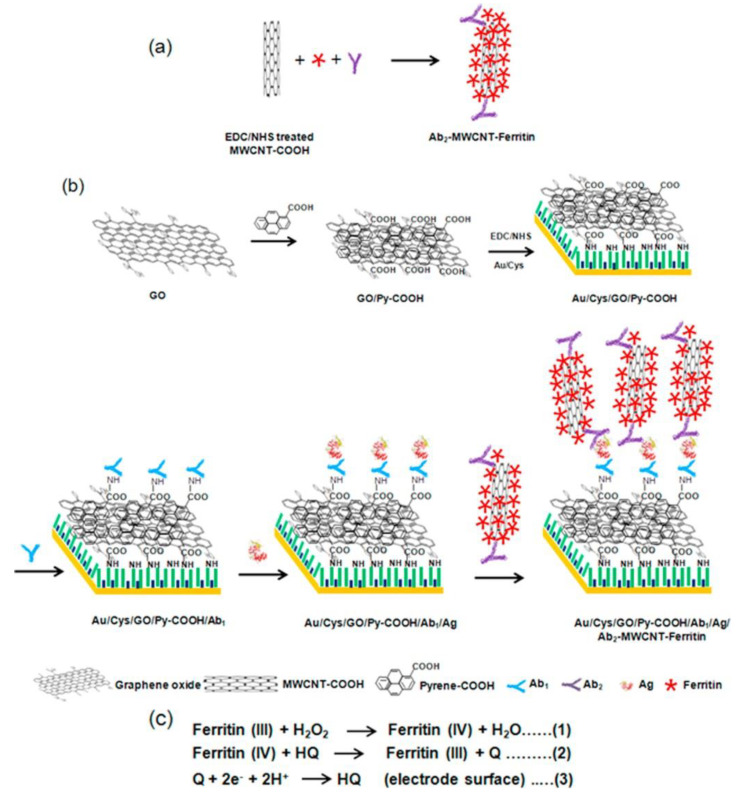
(**a**) Preparation of the Ab_2_ conjugate applied as a signal probe, (**b**) fabrication of the interfacial layer of the immunosensor device, and (**c**) mechanism behind the generation of an electrochemical signal. Reprinted from Ref. [90], Copyright (2016), with permission from Elsevier.

**Figure 7 sensors-20-04022-f007:**
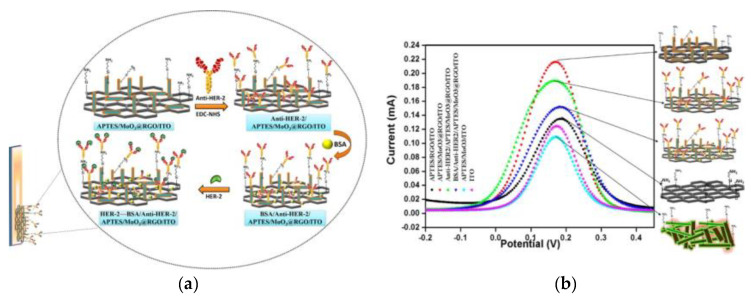
Development of an immunoelectrode for BC biomarker detection (**a**). Electrochemical peak response obtained via differential pulse voltammetry (DPV) at each step of electrode modification (**b**). Reprinted with permission from Ref. [92]. Copyright (2019) American Chemical Society.

**Figure 8 sensors-20-04022-f008:**
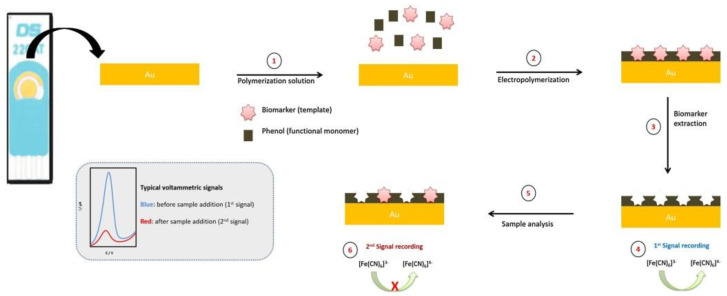
Fabrication and operation principles of the molecularly imprinted polymer-based sensor deposited on a gold screen-printed electrode (AuSPE). Reprinted from Ref. [95], Copyright (2018), with permission from Elsevier.

**Figure 9 sensors-20-04022-f009:**
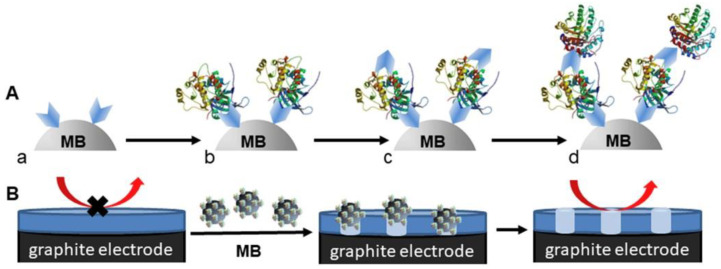
Schematic representation of (**A**) sandwich assembly on magnetic beads (MB) modified (**a**) with either an antibody or an aptamer, (**b**) binding of human epidermal growth factor receptor-2 (HER2)/neu via biorecognition to MB and the further reaction of MB (**c**) with a second antibody or an aptamer and (**d**) binding of the biotinylated cellulase label, through the biotin–streptavidin interaction. The protein structures’ PDB (Protein Data Bank) IDs are: 3PP0 (HER-2/neu; DOI:10.2210/pdb3PP0/pdb) and 4IM4 (cellulase; DOI:10.2210/pdb4IM4/pdb). (**B**) Basic principle of the biosensor operation: electrochemically insulating nitrocellulose film on the surface of porous spectroscopic graphite is digested by MBs only when the analyte is present on the MB by cellulase, and that changes the electrochemical properties of the nitrocellulose-modified graphite surface. Reprinted from Ref. [100], Copyright (2019), with permission from Elsevier.

**Figure 10 sensors-20-04022-f010:**
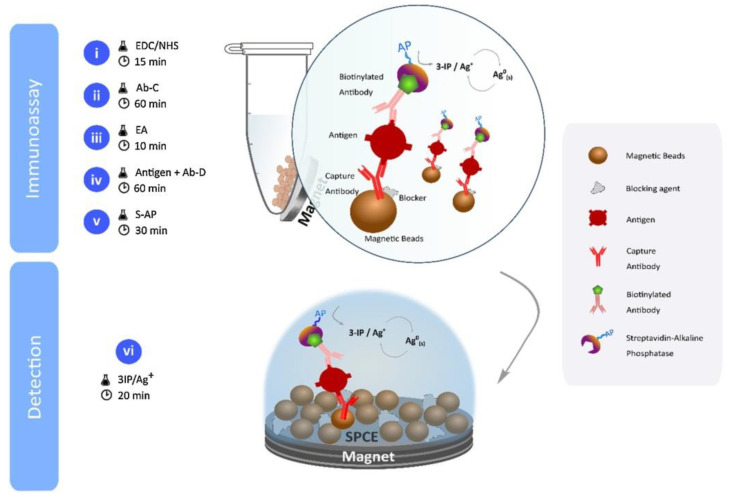
Graphical representation of operation of magnetic bead-based immunoassay. Reprinted from Ref. [97], Copyright (2020), with permission from Elsevier.

**Figure 11 sensors-20-04022-f011:**
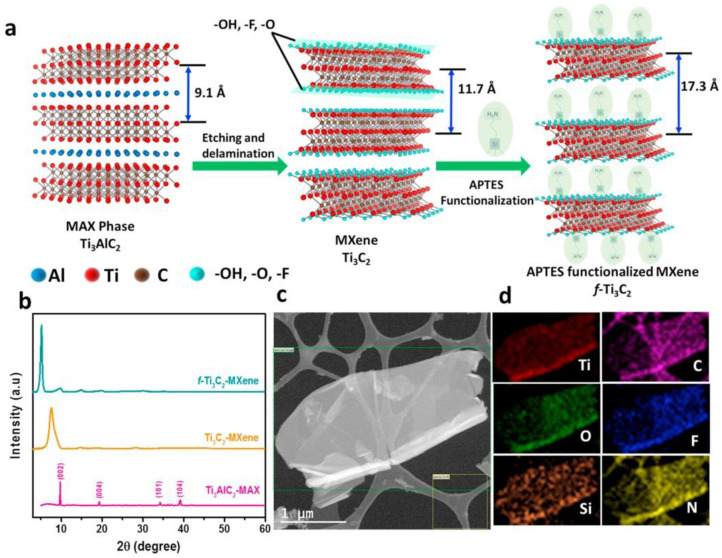
(**a**) Schematic illustration of MXene functionalization. Aluminum layered is etched from the Ti_3_AlC_2_-MAX phase, producing 2D nanosheets terminated with –OH, –O, and –F functional groups. Aminosilane (APTES) is then used to functionalize the MXene surface. (**b**) XRD pattern of the Ti_3_AlC_2_-MAX, MXene and functionalized MXene. (**c**,**d**) TEM image of a single MXene sheet and corresponding elemental map showing the uniform distribution of silicon (Si, brown), oxygen (O, green), and nitrogen (N, dark yellow), and revealing the homogenous functionalization of MXene with APTES. Reprinted from Ref. [113], Copyright 2018, with permission from Elsevier.

**Figure 12 sensors-20-04022-f012:**
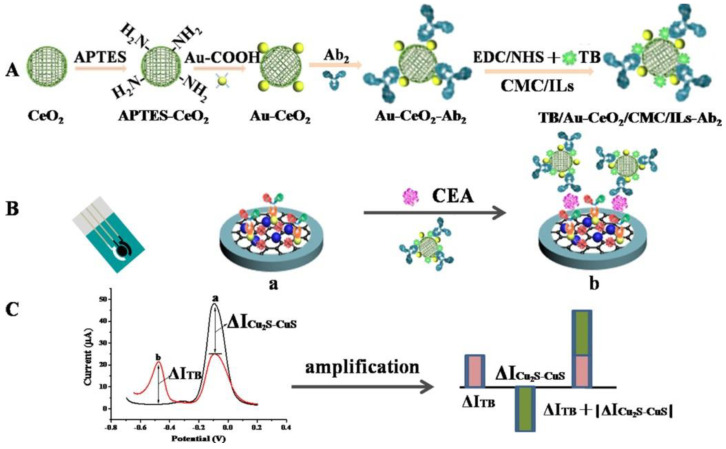
(**A**) Formation of a signal probe. (**B**) Schematic presentation of the immunosensor fabrication. (**C**) A dual-signaling amplification strategy. Reprinted from Ref. [116], Copyright (2018), with permission from Elsevier.

**Figure 13 sensors-20-04022-f013:**
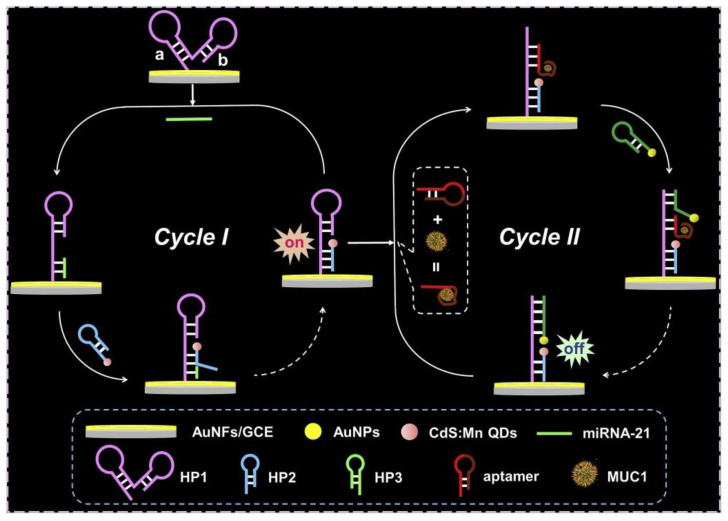
The principle of the fabricated biosensor for the sensitive detection of miRNA-21 and MUC1 based on dual catalytic hairpin assembly. Reprinted from Ref. [119], Copyright 2018, with permission from Elsevier.

**Figure 14 sensors-20-04022-f014:**
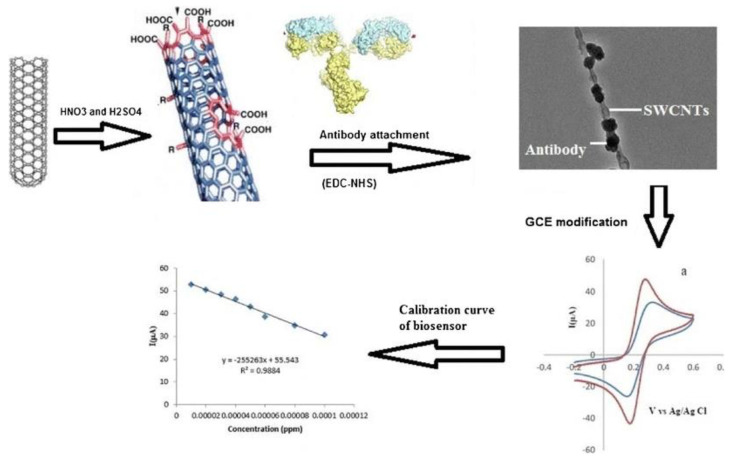
Fabrication of the biosensor for the determination of tissue plasminogen activator. Reprinted by permission from Springer, Ref. [124], Copyright 2018.

**Figure 15 sensors-20-04022-f015:**
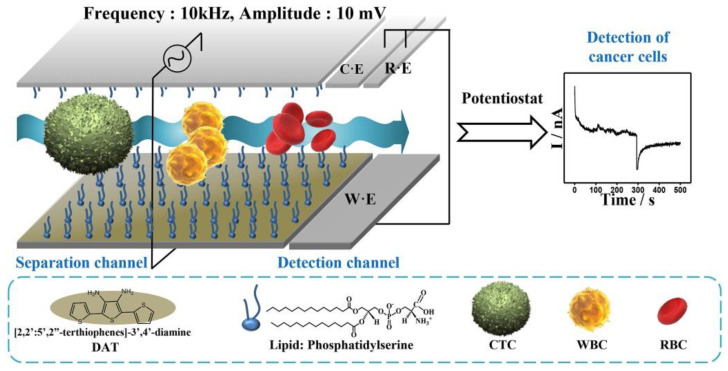
The schematic representation of the design and fabrication of the proposed microfluidic channel. Reprinted from Ref. [128], Copyright 2019, with permission from Elsevier.

**Figure 16 sensors-20-04022-f016:**
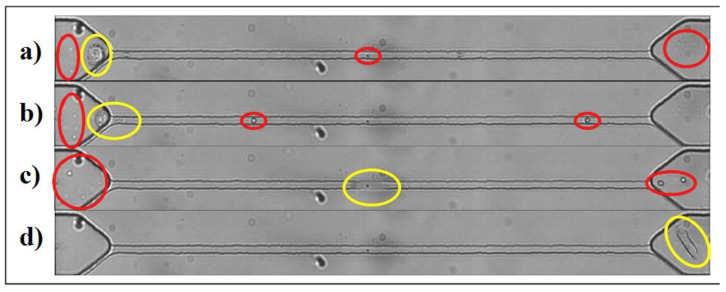
Transit of an MDA-MB-231 cell (circled in yellow) through the constriction region. (**a**) Cancer cell before deformation, (**b**) cell beginning to deform, (**c**) cell in constriction channel, and (**d**) cell after leaving the constriction channel. The surrounding white and red blood cells are indicated by the red circles. Reprinted from Ref. [129], Copyright 2020, with permission from Elsevier.

**Figure 17 sensors-20-04022-f017:**
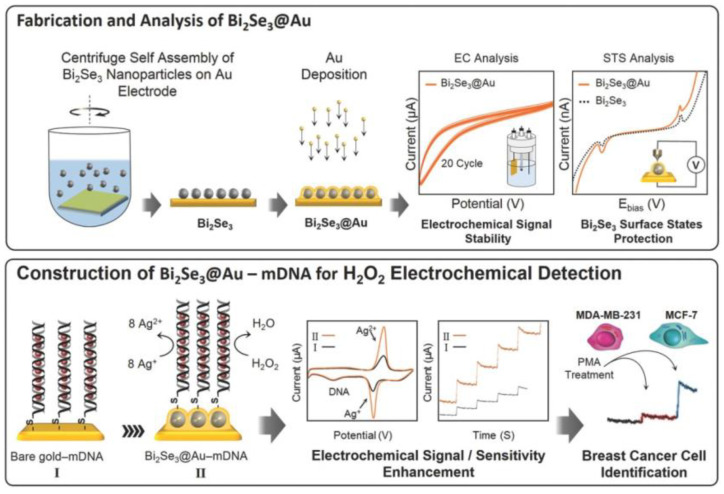
Fabrication process of a Bi_2_Se_3_@Au-DNA electrode, its electrical and electrochemical behavior, and the constitution for the electrochemical detection of H_2_O_2_. Reprinted from Ref. [135], Copyright 2018, with permission from John Wiley and sons.

**Figure 18 sensors-20-04022-f018:**
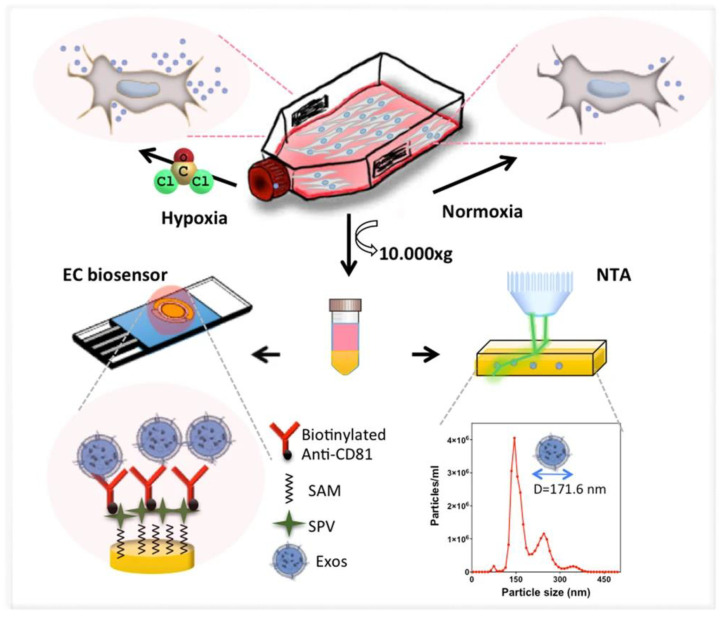
Experimental steps followed throughout the work. MCF-7 cells were exposed to either CoCl_2_-induced hypoxic or normoxic conditions. The isolation of extracellular vesicles (EVs) was done via ultracentrifugation. The characterization and quantification of EVs were done via biosensors that are designed to capture exosomes on the electrode surface by antibodies raised against CD-81 protein expressed on the surface of exosomes. The gold electrode was patterned by thiolated SAM to which streptavidin was covalently immobilized, followed by the attachment of biotinylated anti-CD-81 antibodies. Reprinted with permission from Ref. [142]. Copyright 2018, Nature.

**Figure 19 sensors-20-04022-f019:**
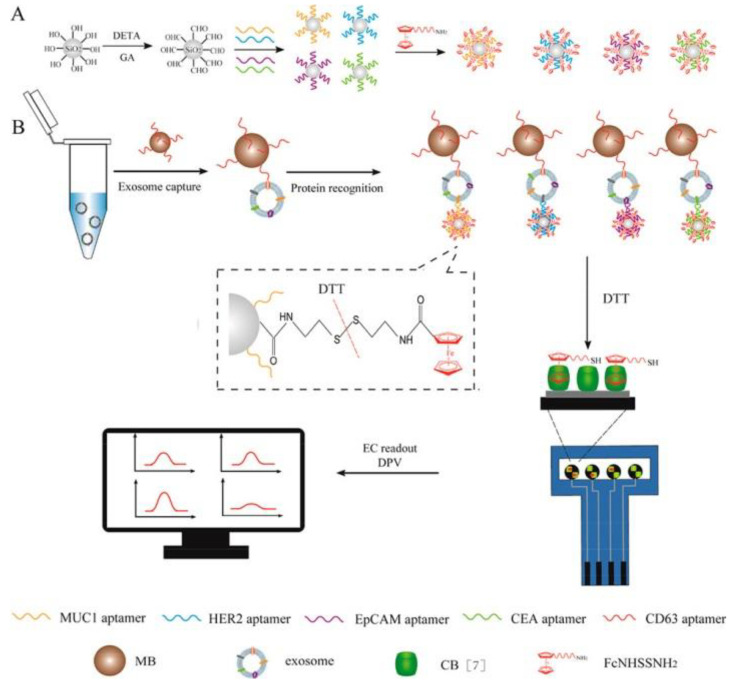
(**A**) Preparation of SiO2 NPs Probes; (**B**) Differential pulse voltammetry (DPV) responses of the magneto-mediated electrochemical sensor for MUC1, HER2, EpCAM, and carcinoembryonic antigen (CEA) markers. Magneto-mediated electrochemical sensor for exosomal proteins analysis based on host–guest recognition. Differential pulse voltammetry (DPV) responses of the magneto-mediated electrochemical sensor for MUC1, HER2, EpCAM, and carcinoembryonic antigen (CEA) markers for the MCF7, SKBR-3, MDA-MB-231, and BT474 cells-derived exosomes at a concentration of 1.2 × 10^6^ particles/μL. Reprinted with permission from reference [143]. Copyright 2020, American Chemical Society.

**Figure 20 sensors-20-04022-f020:**
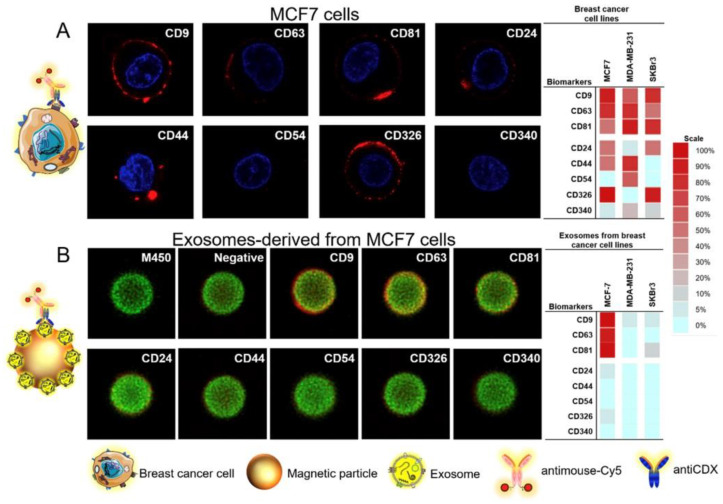
Confocal microscopy study for (**A**) MCF7 breast cancer cell lines and (**B**) their corresponding exosomes covalently immobilized on MPs (exosomes–MPs), followed by indirect labeling with mouse antiCDX antibodies (being CDX either CD9, CD24, CD44, CD54, CD63, CD81, CD326, and CD340 biomarkers) and antimouse-Cy5. The concentration of exosomes was set as 4 × 10^9^
*per* assay. The scale indicates the percentage of positive entities (cells and exosomes-coated MPs in panels A and B, respectively). Reprinted from [144], Copyright 2020, with permission from Elsevier.

**Figure 21 sensors-20-04022-f021:**
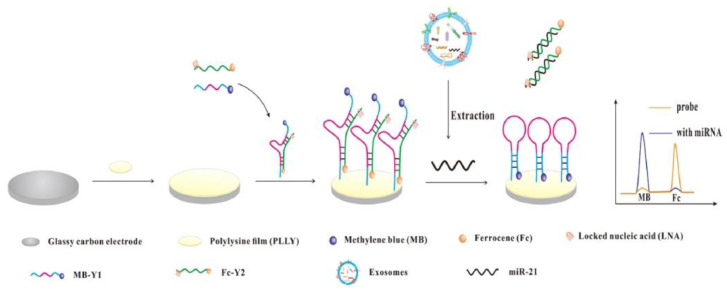
The ratiometric electrochemical biosensor for exosomal miR-21 detection. Reprinted from Ref. [145], Copyright 2020, with permission from Elsevier.

**Table 1 sensors-20-04022-t001:** Candidate breast cancer biomarkers (BRCA1, BRCA2, CA27.29, CA 15-3, CEA, HER-2, VEGF, tPA, CIFRA-21-1, PDGF, OPN).

Biomarker	Size/kDa	Incidence in Cancer	Level in Serum
***BRCA1***	207–220	breast, ovarian, prostate, pancreatic	ND
***BRCA2***	384	Fanconi anemia, breast, ovarian, lung, prostate, pancreatic	ND
**CA27.29**	250–1000	breast	≤37 U/mL
**CA15-3**	290–400	breast	3–30 U/mL
**CEA**	180–200	gastric, pancreatic, lung, breast, medullary thyroid	2–4 ng/mL
**HER-2**	185	breast, ovarian, gastric, prostate	15 ng/mL
**VEGF**	18–27	brain, lung, gastrointestinal, hepatobiliary, renal, breast, ovarian	~220 pg/mL
**TPA**	20–45	breast, lung, pancreatic	109 U/L
**CIFRA-21-1**	40	breast, lung, pancreatic	50 ng/mL
**PDGF**	35	glioblastoma, lung, colorectal, breast, liver and ovarian	(7.5 ± 3.1) ng/mL
**OPN**	41–75	breast, colon, liver, lung, ovarian, prostate	16 ng/mL

**Table 2 sensors-20-04022-t002:** Key characteristics of electrochemical nanobiosensors for the detection of BC biomarkers.

Target Biomarker (Biomolecule)	Bare Electrode	Electrode Modification	Detection	LR	LOD	Refs.
*BRCA 1*	MBCPE	Fe_3_O_4_@Ag, DNA probe	EIS	100 aM–10 nM	30 aM	[63]
	GCE	RGO, MWCNTs, PANHS	CV, EIS	100 aM–10 nM	37 aM	[64]
*PIK3CA* gene	CPE	ssDNA/PIn6COOH/ MoS_2_	CV, EIS	100 aM–10 pM	15 aM	[72]
	GCE	GO, GNR	EIS	15 fM–148 pM	4.5 fM	[77]
	GCE	ZrO_2_-RGO	EIS	10 fM–100 pM	4.3 fM	[78]
MUC MUC	SPCE	CNTs	CV, EIS	0.1–2 U/mL	0.02 U/mL	[83]
	GCE	ferrocene-loaded polymeric micelle	CV	1–1000 cells/mL	10 cells/mL	[85]
	GCE	cDNA-Fc/MXene/Apt/Au	EIS, SWV	1.0 pM–10 mM	0.33 pM	[86]
CA15-3	GE	streptavidin-coated magnetic beads	CV, EIS	ND	15 × 10^−12^ U/mL	[87]
	GE	GO/Py-COOH, MWCNTs	DPV	0.1–20 U/mL	0.01 U/mL	[90]
HER2	ITO	APTES/MoO_3_@RGO	CV, DPV, EIS	0.001–500 ng/mL	0.001 ng/mL (~5.41 fM)	[92]
	GCE	AuNP-ERGO-SWCNTs	EIS	0.1 pg/mL–1 ng/mL	50 fg/mL (~0.27 fM)	[93]
	SPGE	MIP	CV	10–70 ng/mL	1.6 ng/L (~8.65 fM)	[95]
	GE	GNR@Pd SSs—Apt—HRP	EIS	10–200 ng/mL	0.15 ng/mL (~0.81 pM)	[101]
	SPCE	MBs and CdSe@ZnS QDs	DPASV	0.50–50 ng/mL	0.29 ng/mL (~1.57 pM)	[99]
CEA	GCE	aptamer/GLD/CS/ZnS-CdS/MoS_2_	CV	0.05–20 ng/mL	0.031 ng/mL (~0.16 pM)	[106]
	CPE	GNPs and MWCNTs.	CV, EIS	0.4–125 ng/mL	0.09 ng/mL (~0.45 pM)	[107]
	GCE	Au-AgNPs/RGO	CV	0.001–80 ng/mL	0.29 pg/mL (~1.45 fM)	[114]
miRNA-21 and MUC1	GCE	Au nanoflowers	ECL	20 aM–50 pM (miRNA-21) 1 fg mL^−1^–10 ng mL^−1^ (MUC1)	11 aM (miRNA-21) 0.4 fg/mL (~7.27 aM) (MUC1)	[119]
CA 27-29	GCE	Au/MoS_2_/RGO	CV	0.1–100 U/mL	0.08 U/mL	[121]
uPA	FTO	GNS	DPV, CV	1 fM–1 µM	4.8 fM	[122]
tPA	GCE	SWCNTs	CV, EIS	0.1–1.0 ng/mL	0.026 ng/mL (~0.37 pM)	[124]
MCF-7/CTC		RGO/AuNPs/CuO	CV, CA	50–7000 cells/mL	27 cells/mL	[131]
MCF-7	GCE	Au NCs/amino-functionalized MWCNT-NH_2_	CV, EIS	100–1.0 × 10^6^ cells/mL	80 cells/mL	[133]
	GE	Bi_2_Se_3_@Au-mDNA	CV, EIS	100 nM–27 μM	10 nM	[135]
	GCE	Hexagonal carbon nitride tubes	Photo-current	100–1 × 10^5^ cells/mL	17 cells/mL	[137]
	GCE	TiO_2_ nanotubes with graphene	EIS	1000–1 × 10^7^ cells/mL	40 cells/mL	[138]
MDA-MB-231	GE	Non-spherical AuNPs	DPV	10–1 × 10^6^ cells/mL	2 cells/mL	[139]
Cancer stem cells	GE	AgNPs	DPV	10–5 × 10^5^ cells/mL	6 cells/mL	[141]
Exosomes Exosomal miRNA-21	Au SPE	11-MUA	EIS, DPV	10^2^–10^9^ particles/mL	77 particles/mL	[142]
	SPCE	MB SiO_2_ NPs	DPC, EIS	1.2 × 10^3^–1.2 × 10^7^ particles/μL	1.0 × 10^7^ particles/μL	[143]
	m-GEC magnetic	MPs	Ampero-metry	0–1 × 10^6^ particles/μL	10^5^ particles/μL	[144]
	GCE	Polylysine	DPV	10–70 fM	2.3 fM	[145]

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
