# Peer review of "Electrochemical Nanobiosensors for Detection of Breast Cancer Biomarkers"

_sensors, 2020, doi:10.3390/s20144022_

Round 1

Reviewer 1 Report

In this manuscript, the authors reported an approach toward Electrochemical nanobiosensors for detection of breast cancer biomarkers. Generally, current work presents many up to date references, but it is difficult to follow. The manuscript lacks clarity in conveying the overall concept of the review related to nanobiosensor.

  1. The authors should try to emphasize the role of the “nano” dimension in improving the performance of the described biosensors, in order to attract the readership of the journal. The detailed mechanism and problem associated in this direction is not fully covered in this review.
  2. Another suggestion is that the descriptive part can introduce the representative electrochemical detection strategies, while the others references can be simply detailed in the table. The table can be also divided considering criteria relevant for the subject of the review.

Too many fine grained details needs to be skipped – e.g. p15, line 561:

Some phrases, if kept, should also be reformulated:

p1, line 33-34: „Earlier...“

p5, line 196-197: „With...“

p6, line 217-218: „Signal...“

p6, line 221-222: „Beside...“

p7, line 241-243: „Mandli and...“

p9, line 342-344: „Akter...“

Some of the abreviation were already introduced and mentioned again, while another needs to be introduced:

e.g. p12, line 425; line 445

Author Response

In this manuscript, the authors reported an approach toward Electrochemical nanobiosensors for detection of breast cancer biomarkers. Generally, current work presents many up to date references, but it is difficult to follow. The manuscript lacks clarity in conveying the overall concept of the review related to nanobiosensor.

1. The authors should try to emphasize the role of the “nano” dimension in improving the performance of the described biosensors, in order to attract the readership of the journal. The detailed mechanism and problem associated in this direction is not fully covered in this review.

  • The following text was added into the manuscript to describe beneficial role of nanoparticles:

„The speech of the physicist Richard Feynman entitled ‘‘There’s plenty of room at the bottom”, taken place at the Meeting of the American Physical Society in 1959 at CalTech is considered to be the beginning of the nanotechnology era. Significant attention is currently being paid to nanomaterials. Nanomaterials are considered a pivotal tool for numerous applications in part due to their high surface area, compared to their respective bulk forms. Nanostructures with at least one dimension of size of 100 nm (1 nm = 1×10-9 m) or smaller, are extremely useful in a number of areas, such as electronics, aerospace, military, pharmaceuticals, medicine etc. Within last years, there has been an improvement in the synthesis and characterization of different nanomaterials, such as carbon-based nanomaterials, hydrogels, magnetic nanoparticles, metallic nanoparticles, polymer nanoparticles and/or nanocomposites and two dimensional nanomaterials [41, 42].

One of the leading areas of the state-of-the-art nanoscience and nanotechnology presents development of 1D (lamellar), 2D (fibers) and 3D (particulates) nanomaterials/nanoparticles for application in biosensors.

The application of nanomaterials to design (bio)sensing platforms offers exceptional electronic, magnetic, mechanical, and optical properties. Nanomaterials can increase the surface of the transducing area of the sensors which in turn provides enhanced catalytic activity. Electroactive properties of nanoparticles towards certain reactions have been widely exploited in biosensing applications. Nanometer-size structures have large surface-to-volume ratio, controlled morphology and structure that would scale down the characteristic size, a clear advantage when the sample volume is critical. The integration of advanced 2D nanomaterial MXene into biosensors architecture brings advantage of hydrophilic character due to functional groups onto nanoscale surface [43]. However, advances in nanomaterial biofunctionalization are crucial to achieve higher specificity in biosensing. To that end, nanomaterials can be “decorated” with different (bio)receptors offering specific recognition for biosensing [44-49]. There are basically two approaches applied to design nanobiosensors i.e. application of nanoparticles for modification of electrode surfaces (Approach 1, Scheme 1), or application of nanoparticles to make signal nanoprobes enhancing generated signal (Approach 2, Scheme 1). There are some nanobiosensors constructed using both amplification approaches (hybrid biosensing, i.e. Approach 3). In the forthcoming sections when discussing particular nanobiosensors also amplification strategies are indicated.”

2. Another suggestion is that the descriptive part can introduce the representative electrochemical detection strategies, while the others references can be simply detailed in the table. The table can be also divided considering criteria relevant for the subject of the review.

  • We prepared Scheme 1 to show how nanoparticles can amplify electrochemical signal either by patterning of the electrode (Approach 1), by preparing a signal probe (Approach 2) or by combining both approaches (Approach 3). In the sub-sections in the manuscript, when discussing particular nanobiosensors also amplification strategies are indicated. Moreover, Table 2 was revised.

Too many fine grained details needs to be skipped – e.g. p15, line 561:

Some phrases, if kept, should also be reformulated:

p1, line 33-34: „Earlier...“

p5, line 196-197: „With...“

p6, line 217-218: „Signal...“

p6, line 221-222: „Beside...“

p7, line 241-243: „Mandli and...“

p9, line 342-344: „Akter...“

  • The phrases were reformulated, as it was required.

Some of the abbreviation were already introduced and mentioned again, while another needs to be introduced:

e.g. p12, line 425; line 445

  • The abbreviations in the whole text were controlled, as it was required.

Reviewer 2 Report

The review entitled “Electrochemical nanobiosensors for detection of breast cancer biomarker” from Gajdosova et al. as reported the recent advances in electrochemical biosensors for breast cancer detection.

For sure the topic is interesting, even if different reviews have been presented in the last years. An update is always useful.

The language and style are not always fine, and it must be improved, especially in some parts that are worst than others. A few misprints and some parts that must be rephrased to be understandable require a revision. In some part of the paper hyper-links to wikipedia are still present in the text. This is for sure not a good index of quality...In other parts of the text (e.g. caption of Fig 11) the text is the same of the reference.

In general, the introduction (par. 1) and the following introductory paragraphs (2 and 3) are more or less fine. From page 4 (par. 4) the quality drops. Apart from the first lines of this last paragraph, in the different sections there is a list of different papers that reports the development of many sensors, but without a real critical view of these works. They seems to be just listed to increase the number of lines of the paper, and apparently sometimes without a logical order or classification. Please, in a review you should critically analyse the state of the art, not only report some papers without linking them.

At the same time, a real focus on challenges and future prespectives is missing or really hidden in the text. The authors stated in the introduction “There are only two review papers specifically covering electrochemical biosensing of breast cancer biomarkers published in 2017 [17, 18], but with only a minor coverage of beneficial properties of nanoparticles within electrochemical transducing schemes.” but they do not highlighted the advantages or disadvantages of such approach. So, it is not really clear in the text if the presence of nanoparticles or nanomaterials really helps the electrochemical biosensing of breast cancer biomarkers.

Therefore, it seems that this work as the potentiality to became a nice review, and I think the paper can be published in Sensors because it perfectly fit the aim of the journal, but in order to improve its overall quality, I listed a few points the authors should address before acceptance:

- A few misprints and mistakes are present, so I suggest to read the paper carefully and proofread it. Maybe a native speaker is required. A Few examples (not exaustive!!! there are many more): lines 223, 425, 529, 558, 560, 669, 702, 780...etc

- Take care with acronym. They should be defined the first time you insert them in the manuscript. Please, check the whole paper, because sometimes they are defined not the first time you use it. At the same time, you should take care because you used a lot of different acronyms and sometimes you only used it once or twice. This should be avoided: it is not easy to remember through the paper all of them, especially if these are not common or if you used similar acronyms for different things. Another example is for instance at line 522 “molybdenum disulfide (MoS2 )”: people that know Chemistry use this nomenclature frequently, so it is not required to specify the acronyms. What is TB at line 559? Maybe Toluidine Blue O that was not defined? Since you used only there, you should avoid the acronym.

- There are many different kind of electrochemical biosensors. It should be useful to add in the introduction a brief paragraph with a short overview of the main class of electrochemical sensors. Maybe a scheme or a figure can help you in this.

- Most of the images show low resolution and are too small to be readable. Especially Fig. 4, 5, 6, 7, 8, 10, 11, 13, 16, 17, 18, 19. Please, revise them.

- Line 315: “to resist non specific interactions” please revise it.

- Lines 328-330. You already described mucins in the previous paragraph (lines 291-293). You should move there part of this description, and maybe you can combine paragraphs 4.3 and 4.4.

- In lines 415-424 two examples from Soares et al are reported. Neither in this case the authors critically commented the results. So which is the best option?

- Figure captions of Fig 9 and 17 should be revised because it is quite difficult to understand what the authors mean. Please rephrase these captions.

- Line 524. “to develop the aptasensor”. The authors are describing a paper from Wang et al (since line 521) at only after a while they introduce the aptasensor. Please revise it.

- At line 560, probably you miss something, because carboxymethyl chitosan has primary amine, not NH…

- Caption of Fig. 11 si completely copied from the source (ref 113).

- Lines 731-733, the authors already introduced MCF7 cell line, so please this three line must be moved from here to previous section of the paper.

- Lines 736-746 and 750-751 are not clear. Please, revise it to make these lines understandable.

- Lines 780 and 785 are very similar…

- Lines 794-812. This part is a bit confused and should be revised.

- Caption of Fig. 19: “with mouse antiCDX...” antibodies? Aptamers? Please, revise it.

- Line 879: “MB”?

- In line 880 ferrocene is abbreviated “Fc” but this can be also the “Fc fragment of an antibody” I suggest to avoid this acronyms for a more clear text.

- Table 2 is a nice idea to summarize the different sensors described in the text. Anyway, I suggest to better organize it. At least, in the first column the cells containing the same biomarker should be merged, so it will be more readable. The LOD concentration should be possibly in international standard units; so fM or aM is fine, but the values expressed as ug/mL or something else should be appropriately converted. At least the same unit of measurement should be used for the same biomarkers to allow a fruitful comparison.

- Line 927. Please revise it: it is not clear.

- SWCNTs in abbreviations is only Single-walled...please revise it.

Author Response

The review entitled “Electrochemical nanobiosensors for detection of breast cancer biomarker” from Gajdosova et al. as reported the recent advances in electrochemical biosensors for breast cancer detection.

For sure the topic is interesting, even if different reviews have been presented in the last years. An update is always useful.

The language and style are not always fine, and it must be improved, especially in some parts that are worst than others. A few misprints and some parts that must be rephrased to be understandable require a revision. In some part of the paper hyper-links to wikipedia are still present in the text. This is for sure not a good index of quality...In other parts of the text (e.g. caption of Fig 11) the text is the same of the reference.

  • The misprints and hyper-links were corrected, as it was required.

In general, the introduction (par. 1) and the following introductory paragraphs (2 and 3) are more or less fine. From page 4 (par. 4) the quality drops. Apart from the first lines of this last paragraph, in the different sections there is a list of different papers that reports the development of many sensors, but without a real critical view of these works. They seems to be just listed to increase the number of lines of the paper, and apparently sometimes without a logical order or classification. Please, in a review you should critically analyze the state of the art, not only report some papers without linking them.

  • A Scheme 1 was introduced in the manuscript to show how nanoparticles can amplify electrochemical signal either by patterning of the electrode (Approach 1), by preparing a signal probe (Approach 2) or by combining both approaches (Approach 3). In sub-sections in the manuscript, when discussing particular nanobiosensors also amplification strategies are indicated. Moreover, for detection of each cancer biomarker the text is organized in a way to discuss Approach 1-based nanobiosensors, followed by Approach 2-based nanobiosensors and followed by approach 3-based nanobiosensors.

At the same time, a real focus on challenges and future perspectives is missing or really hidden in the text. The authors stated in the introduction “There are only two review papers specifically covering electrochemical biosensing of breast cancer biomarkers published in 2017 [17, 18], but with only a minor coverage of beneficial properties of nanoparticles within electrochemical transducing schemes.” but they do not highlighted the advantages or disadvantages of such approach. So, it is not really clear in the text if the presence of nanoparticles or nanomaterials really helps the electrochemical biosensing of breast cancer biomarkers.

The paragraph related to importance and beneficial advantages of nanomaterials/nanoparticles applying in biosensor, schematic illustration of different approaches for their application were included, as it was advised. The following text was added into the manuscript:

„The speech of the physicist Richard Feynman entitled ‘‘There’s plenty of room at the bottom”, taken place at the Meeting of the American Physical Society in 1959 at CalTech is considered to be the beginning of the nanotechnology era. Significant attention is currently being paid to nanomaterials. Nanomaterials are considered a pivotal tool for numerous applications in part due to their high surface area, compared to their respective bulk forms. Nanostructures with at least one dimension of size of 100 nm (1 nm = 1×10-9 m) or smaller, are extremely useful in a number of areas, such as electronics, aerospace, military, pharmaceuticals, medicine etc. Within last years, there has been an improvement in the synthesis and characterization of different nanomaterials, such as carbon-based nanomaterials, hydrogels, magnetic nanoparticles, metallic nanoparticles, polymer nanoparticles and/or nanocomposites and two dimensional nanomaterials [41, 42].

One of the leading areas of the state-of-the-art nanoscience and nanotechnology presents development of 1D (lamellar), 2D (fibers) and 3D (particulates) nanomaterials/nanoparticles for application in biosensors.

The application of nanomaterials to design (bio)sensing platforms offers exceptional electronic, magnetic, mechanical, and optical properties. Nanomaterials can increase the surface of the transducing area of the sensors which in turn provides enhanced catalytic activity. Electroactive properties of nanoparticles towards certain reactions have been widely exploited in biosensing applications. Nanometer-size structures have large surface-to-volume ratio, controlled morphology and structure that would scale down the characteristic size, a clear advantage when the sample volume is critical. The integration of advanced 2D nanomaterial MXene into biosensors architecture brings advantage of hydrophilic character due to functional groups onto nanoscale surface [43]. However, advances in nanomaterial biofunctionalization are crucial to achieve higher specificity in biosensing. To that end, nanomaterials can be “decorated” with different (bio)receptors offering specific recognition for biosensing [44-49]. There are basically two approaches applied to design nanobiosensors i.e. application of nanoparticles for modification of electrode surfaces (Approach 1, Scheme 1), or application of nanoparticles to make signal nanoprobes enhancing generated signal (Approach 2, Scheme 1). There are some nanobiosensors constructed using both amplification approaches (hybrid biosensing, i.e. Approach 3). In the forthcoming sections when discussing particular nanobiosensors also amplification strategies are indicated.”

Therefore, it seems that this work as the potentiality to became a nice review, and I think the paper can be published in Sensors because it perfectly fit the aim of the journal, but in order to improve its overall quality, I listed a few points the authors should address before acceptance:

- A few misprints and mistakes are present, so I suggest to read the paper carefully and proofread it. Maybe a native speaker is required. A Few examples (not exaustive!!! there are many more): lines 223, 425, 529, 558, 560, 669, 702, 780...etc

  • The misprints and mistakes were corrected, as it was required.

- Take care with acronym. They should be defined the first time you insert them in the manuscript. Please, check the whole paper, because sometimes they are defined not the first time you use it. At the same time, you should take care because you used a lot of different acronyms and sometimes you only used it once or twice. This should be avoided: it is not easy to remember through the paper all of them, especially if these are not common or if you used similar acronyms for different things. Another example is for instance at line 522 “molybdenum disulfide (MoS2 )”: people that know Chemistry use this nomenclature frequently, so it is not required to specify the acronyms. What is TB at line 559? Maybe Toluidine Blue O that was not defined? Since you used only there, you should avoid the acronym.

  • TB presents toluidine blue, it was included, as it was requested.
  • The acronyms were checked and added in the whole text, as it was required.

- There are many different kind of electrochemical biosensors. It should be useful to add in the introduction a brief paragraph with a short overview of the main class of electrochemical sensors. Maybe a scheme or a figure can help you in this.

  • We prepared Scheme 1 to show how nanoparticles can amplify electrochemical signal either by patterning of the electrode (Approach 1), by preparing a signal probe (Approach 2) or by combining both approaches (Approach 3). In sub-sections in the manuscript, when discussing particular nanobiosensors also amplification strategies are indicated.

- Most of the images show low resolution and are too small to be readable. Especially Fig. 4, 5, 6, 7, 8, 10, 11, 13, 16, 17, 18, 19. Please, revise them.

  • All images were downloaded and added with the highest resolution possible.

- Line 315: “to resist non specific interactions” please revise it.

  • It was revised, as it was required.

- Lines 328-330. You already described mucins in the previous paragraph (lines 291-293). You should move there part of this description, and maybe you can combine paragraphs 4.3 and 4.4.

  • CA 15-3 biomarker presents a soluble form of mucin, so we marked this section as sub-chapter with published papers related to CA 15.3 detection to paragraph 4.3.

- In lines 415-424 two examples from Soares et al are reported. Neither in this case the authors critically commented the results. So which is the best option?

  • The second immobilization approach containing Cys presents better option and showed higher sensitivity. It was included, as it was required.

- Figure captions of  Fig 9 and 17 should be revised because it is quite difficult to understand what the authors mean. Please rephrase these captions.

  • Both figure captions were rephrased to make them clearer.

- Line 524. “to develop the aptasensor”. The authors are describing a paper from Wang et al (since line 521) at only after a while they introduce the aptasensor. Please revise it.

  • It was revised, as it was required.

- At line 560, probably you miss something, because carboxymethyl chitosan has primary amine, not NH…

  • The sentence was corrected.

- Caption of Fig. 11 is completely copied from the source (ref 113).

  • Caption to this Figure is rewritten.

- Lines 731-733, the authors already introduced MCF7 cell line, so please this three line must be moved from here to previous section of the paper.

  • These lines were moved, as it was required.

- Lines 736-746 and 750-751 are not clear. Please, revise it to make these lines understandable.

  • Lines 736-746 and 750-751 were revised, as it was

- Lines 780 and 785 are very similar…

  • Lines 780 and 785 were revised, as it was required.

- Lines 794-812. This part is a bit confused and should be revised.

  • This part was revised, as it was required.

- Caption of Fig. 19: “with mouse antiCDX...” antibodies? Aptamers? Please, revise it.

  • Caption of Fig. 19 was corrected “with mouse antiCDX antibodies”, as it was required.

- Line 879: “MB”?

  • MB presents abbreviation for methylene blue, that is used in the text with full expression to differentiate it from MB, which is used for magnetic beads..

- In line 880 ferrocene is abbreviated “Fc” but this can be also the “Fc fragment of an antibody” I suggest to avoid this acronyms for a more clear text.

  • The abbreviation for ferrocene Fc is not included, instead full name is used.

- Table 2 is a nice idea to summarize the different sensors described in the text. Anyway, I suggest to better organize it. At least, in the first column the cells containing the same biomarker should be merged, so it will be more readable. The LOD concentration should be possibly in international standard units; so fM or aM is fine, but the values expressed as ug/mL or something else should be appropriately converted. At least the same unit of measurement should be used for the same biomarkers to allow a fruitful comparison.

  • Table 2 was revised, first column of the cells was merged and LODs in international units were completed in the cases of biomarkers, when it was possible.

- Line 927. Please revise it: it is not clear.

- SWCNTs in abbreviations is only Single-walled...please revise it.

  • The Line 927 and abbreviation were revised, as it was required.

Reviewer 3 Report

The paper describes the aaplication of electrochemical biosensors constructed with nanomaterial to detect breast cancer biomarkers.
The authors grouped the described biosensors according to the biomarkers detected. There is a satisfactory volume of references discussed. The paper focus a current topic and could be acessed by interested readers.
However, the authors mention in lines 58 and 59; “but with only a minor coverage of beneficial properties of nanoparticles within electrochemical transducing schemes”. I think that this part is missing in the text. There are many biosensors constructed with nanoparticle describe in the text, but the advantanges in relation to other material and the interesting properties of the nanomaterial for constructed biosensor is lacking. This point is very important, considering that is a differential of this review.
In addition, there is a need for more careful correction of some typos. For example:
line 347: anorganic acids
line 422: immoblisation

Author Response

The paper describes the aplication of electrochemical biosensors constructed with nanomaterial to detect breast cancer biomarkers.

The authors grouped the described biosensors according to the biomarkers detected. There is a satisfactory volume of references discussed. The paper focus a current topic and could be acessed by interested readers.
However, the authors mention in lines 58 and 59; “but with only a minor coverage of beneficial properties of nanoparticles within electrochemical transducing schemes”. I think that this part is missing in the text. There are many biosensors constructed with nanoparticle describe in the text, but the advantanges in relation to other material and the interesting properties of the nanomaterial for constructed biosensor is lacking. This point is very important, considering that is a differential of this review. 

  • „The speech of the physicist Richard Feynman entitled ‘‘There’s plenty of room at the bottom”, taken place at the Meeting of the American Physical Society in 1959 at CalTech is considered to be the beginning of the nanotechnology era. Significant attention is currently being paid to nanomaterials. Nanomaterials are considered a pivotal tool for numerous applications in part due to their high surface area, compared to their respective bulk forms. Nanostructures with at least one dimension of size of 100 nm (1 nm = 1×10-9 m) or smaller, are extremely useful in a number of areas, such as electronics, aerospace, military, pharmaceuticals, medicine etc. Within last years, there has been an improvement in the synthesis and characterization of different nanomaterials, such as carbon-based nanomaterials, hydrogels, magnetic nanoparticles, metallic nanoparticles, polymer nanoparticles and/or nanocomposites and two dimensional nanomaterials [41, 42].
  • One of the leading areas of the state-of-the-art nanoscience and nanotechnology presents development of 1D (lamellar), 2D (fibers) and 3D (particulates) nanomaterials/nanoparticles for application in biosensors.
  • The application of nanomaterials to design (bio)sensing platforms offers exceptional electronic, magnetic, mechanical, and optical properties. Nanomaterials can increase the surface of the transducing area of the sensors which in turn provides enhanced catalytic activity. Electroactive properties of nanoparticles towards certain reactions have been widely exploited in biosensing applications. Nanometer-size structures have large surface-to-volume ratio, controlled morphology and structure that would scale down the characteristic size, a clear advantage when the sample volume is critical. The integration of advanced 2D nanomaterial MXene into biosensors architecture brings advantage of hydrophilic character due to functional groups onto nanoscale surface [43]. However, advances in nanomaterial biofunctionalization are crucial to achieve higher specificity in biosensing. To that end, nanomaterials can be “decorated” with different (bio)receptors offering specific recognition for biosensing [44-49]. There are basically two approaches applied to design nanobiosensors i.e. application of nanoparticles for modification of electrode surfaces (Approach 1, Scheme 1), or application of nanoparticles to make signal nanoprobes enhancing generated signal (Approach 2, Scheme 1). There are some nanobiosensors constructed using both amplification approaches (hybrid biosensing, i.e. Approach 3). In the forthcoming sections when discussing particular nanobiosensors also amplification strategies are indicated.”

Moreover, we prepared Scheme 1 to show how nanoparticles can amplify electrochemical signal either by patterning of the electrode (Approach 1), by preparing a signal probe (Approach 2) or by combining both approaches (Approach 3). In sub-sections in the manuscript, when discussing particular nanobiosensors also amplification strategies are indicated.

In addition, there is a need for more careful correction of some typos. For example:
line 347: anorganic acids
line 422: immoblisation

  • The typos were corrected, as it was required.

Round 2

Reviewer 1 Report

The article seems better structured. But due too many tracked changes is difficult to be properly read. It would be useful the submission of a second file, as a second corrected version - For instance, Figure 9 has in this version no capture.

Some English minor corrections are still needed (e.g. Line 1428: identified)

Line 1353:  methylene blue (here in my opinion the abbreviation should be kept)

Author Response

The article seems better structured. But due too many tracked changes is difficult to be properly read. It would be useful the submission of a second file, as a second corrected version - For instance, Figure 9 has in this version no capture.

R: We are sorry that the version of the manuscript with track changes was difficult to read. In the clean revised version of the manuscript, there is figure caption to Fig. 9 present.

Some English minor corrections are still needed (e.g. Line 1428: identified)

R: We carefully checked and corrected all typos in the revised version of the manuscript.

Line 1353:  methylene blue (here in my opinion the abbreviation should be kept).

R: We would like to keep expression “methylene blue” without any abbreviation since abbreviation of MB is use throughout the manuscript for magnetic beads. Moreover a term “magnetic beads” is used more often in the manuscript, when compared to a term “methylene blue”.    

Reviewer 2 Report

First of all, I want to thank the authors because they mostly accomplished all the requests in the revised version of the manuscript. I think that now the paper is more or less ready to be published in Sensors. I only have a few minor concerns:
-in Scheme 1, please, instead of “nanoparticles” I suggest you to use “nanomaterials” because sheet of 2D materials are not really nanoparticles.
- line 464. What about “Approach 3”? Could I miss something? In Scheme 1 there are only two approaches. I noticed you reply that the third approach is the combination of 1 and 2, but this is not clear to the reader. You should state it clearly at least in caption of Scheme1. At the same time, ok for the order you choose to list the different works you cited, but maybe this should be a little bit more clear to the reader. So, it is ok that you report the papers grouped following approach 1, then 2, then 3. Did you think that this order is also linked to the type of biomarker? If there is this trend (as it seems by your order) you should state it. This can be a critically relevant comment.
- part of lines 511-513 should be moved to the previous subsection. You already introduced MUC protein earlier in the text....
- in par 4.5, you removed the sentence “HER2 (185 kDa) belongs to the human epidermal growth factor receptor family.” but this is important to introduce what is HER2 to not biochemist audience. So I suggest you to keep the sentence, maybe by adding a reference from the literature and not from wikipedia.
- The authors moved different parts of the paper, and changed many parts. Maybe they miss caption of new Figure 9?
- As already said, you should not define molybdenum disulfide (MoS2) (line 805)
- on page 36, “cells per mL” should be “cells/mL”

Author Response

First of all, I want to thank the authors because they mostly accomplished all the requests in the revised version of the manuscript. I think that now the paper is more or less ready to be published in Sensors. I only have a few minor concerns:

-in Scheme 1, please, instead of “nanoparticles” I suggest you to use “nanomaterials” because sheet of 2D materials are not really nanoparticles.

R: A term “nanoparticles” was replaced by a term “nanomaterials” in Scheme 1 as suggested by the Reviewer.

- line 464. What about “Approach 3”? Could I miss something? In Scheme 1 there are only two approaches. I noticed you reply that the third approach is the combination of 1 and 2, but this is not clear to the reader. You should state it clearly at least in caption of Scheme1. At the same time, ok for the order you choose to list the different works you cited, but maybe this should be a little bit more clear to the reader. So, it is ok that you report the papers grouped following approach 1, then 2, then 3. Did you think that this order is also linked to the type of biomarker? If there is this trend (as it seems by your order) you should state it. This can be a critically relevant comment.

R: Caption to Scheme 1 was completed to describe Approach 3 discussed in the text. A revised version of the caption to Scheme 1 can be now read as follows: “Please note than we recognize Approach 3 applied to design biosensor devices by combination of the nanomaterial/nanoparticle modified electrode (Approach 1) with use of a signal nanoprobe (Approach 2) within one biosensor device.“

Relevant part of the Conclusions was updated and can be read as follows: “This review provides evidence that electrochemical detection principles can offer ultrasensitive detection platforms for detection of various BC biomarkers with LOD in some cases down to single molecule level thank to use of a wide range of nanoparticles (Table 2). Such biosensors discussed in this review were mainly based on modification of the electrodes by nanomaterials (i.e. Approach 1; 65%), followed by design of signal probes based on nanomaterials (i.e. Approach 2; 22%) with design of a hybrid approach using nanomaterials for modification of electrodes and for design of signal probes (i.e. Approach 3; 13%).“

- part of lines 511-513 should be moved to the previous subsection. You already introduced MUC protein earlier in the text....

R: The section dealing with detection of CA15-3 is now part of the Section “4.3. Detection of mucins”, as suggested by the Reviewer.

- in par 4.5, you removed the sentence “HER2 (185 kDa) belongs to the human epidermal growth factor receptor family.” but this is important to introduce what is HER2 to not biochemist audience. So I suggest you to keep the sentence, maybe by adding a reference from the literature and not from wikipedia.

R: The following sentence was added at the beginning of Section 4.4: “HER2 (185 kDa) i.e. human epidermal growth factor receptor belongs to a family of receptor tyrosine kinases family [91].“

- The authors moved different parts of the paper, and changed many parts. Maybe they miss caption of new Figure 9?

R: caption to Fig. 9 is present in the revised version of the manuscript.

- As already said, you should not define molybdenum disulfide (MoS2) (line 805)

R: The sentence was updated and can be read as follows: “Wang et al. developed a label-free aptasensor based on electrochemiluminescent (ECL) strategy with ZnS-CdS NP-decorated molybdenum disulfide (MoS2, a 2D nanomaterial [106]) nanocomposite for CEA detection [107].“

- on page 36, “cells per mL” should be “cells/mL”

R: All such expressions in relation to cells, particles or exosomes were corrected in the revised version of the manuscript as suggested by the Reviewer.